# Manipulating microRNA *miR408* enhances both biomass yield and saccharification efficiency in poplar

Yayu Guo [1,13], Shufang Wang [1,2,13], Keji Yu[1], Hou-Ling Wang [1], Huimin Xu [1,3], Chengwei Song [1,4], Yuanyuan Zhao [1], Jialong Wen [5], Chunxiang Fu [6], Yu Li[6], Shuizhong Wang [5], Xi Zhang [1], Yan Zhang [1], Yuan Cao [7], Fenjuan Shao[7], Xiaohua Wang[2], Xin Deng [2], Tong Chen [2], Qiao Zhao[8], Lei Li [9], Guodong Wang [10], Paul Grünhofer [11], Lukas Schreiber [11], Yue Li[1], Guoyong Song [5], Richard A. Dixon [1,12,14] ✉ & Jinxing Lin [1,14] ✉

The conversion of lignocellulosic feedstocks to fermentable sugar for biofuel production is inefficient, and most strategies to enhance efficiency directly target lignin biosynthesis, with associated negative growth impacts. Here we demonstrate, for both laboratory- and field-grown plants, that expression of *Pag-miR408* in poplar (*Populus alba × P. glandulosa*) significantly enhances saccharification, with no requirement for acid-pretreatment, while promoting plant growth. The overexpression plants show increased accessibility of cell walls to cellulase and scaffoldin cellulose-binding modules. Conversely, *Pag-miR408* loss-of-function poplar shows decreased cell wall accessibility. Over-expression of *Pag-miR408* targets three *Pag-LACCASES*, delays lignification, and modestly reduces lignin content, S/G ratio and degree of lignin poly-merization. Meanwhile, the *LACCASE* loss of function mutants exhibit sig-nificantly increased growth and cell wall accessibility in xylem. Our study shows how *Pag-miR408* regulates lignification and secondary growth, and suggest an effective approach towards enhancing biomass yield and sacchar-ification efficiency in a major bioenergy crop.

Plant cells walls are the primary source of terrestrial biomass and lig-nocellulosic biomass has been considered an important source of simple sugars for downstream conversion. Crop traits can be opti-mized by regulating single genes to achieve significantly increased yield or stress resistance[1–5]. Moreover, engineering plants cell walls for more efficient deconstruction is becoming a central technology in the generation of biofuels and bio-based chemicals and materials from lignocellulosic biomass[6–9]. The approach usually targets structural genes involved in the biosynthesis of cell wall polymers, the most common target being the lignin polymer[10,11]. However, genetically engineered low-lignin plants are often weak, stunted and produce less total biomass[12–14]. This may be due to lack of lignin deposition in critical

cells such as xylem, tracheary elements, and vessels[15], altered bio-synthesis of related secondary metabolites (flavonoids, coumarins, and lignans) that might influence plant growth and development[13,16], or ectopic activation of defense responses with associated growth trade-offs[14,17].

Approaches to overcoming the growth defects in lignin-modified plants have included screening for suppressor mutations[13], blocking defense signaling pathways[17], restoring lignification to critical vascular tissues[15,18], and generating haplo-insufficient alleles through gene editing[15]. In these cases, the resulting plants still exhibit small yield penalties. Techno-economic analysis reveals that yield, rather than cell wall compositional parameters, is the critical factor in the economic

viability of a bioenergy crop such as poplar (*Populus* spp)[19]. There is therefore a need to develop strategies for biomass engineering that can combine reduced cell wall recalcitrance with enhanced yield. One approach is to harness genetic changes that pleiotropically affect secondary growth parameters.

MicroRNAs (miRNA), endogenous small noncoding RNAs of 21–24 nucleotides in length, are key eukaryotic gene regulators that play critical roles in plant development and stress tolerance[20], and some have targets that may be involved in secondary wall formation and lignification[21,22]. *miR408* is a highly conserved miRNA of 21 nucleotides, first identified in *Arabidopsis thaliana*[23]. Its over-expression increases biomass and seed yield in *Arabidopsis* and rice, potentially through effects on the copper-containing proteins plantacyanin and laccase[24]. These growth effects, coupled with the suggestion of laccase as a target, suggest the possibility that manipulating *miR408* might overcome the growth defects caused by the down-regulation of lignin synthesis through direct targeting of structural genes.

Here, we show that overexpression of *Pag-miR408* in hybrid poplar targets three *LACCASES*, co-down-regulation of which can significantly improve plant growth but also increase cell wall saccharification efficiency without acid pretreatment, as confirmed by using genome editing to simultaneously knock out all three targets of *Pag-miR408*. Although the associated changes in lignin properties as a result of *Pag-miR408* overexpression are only modest, the large increase in saccharification efficiency is associated with changes in xylem development that may lead to increased accessibility of cell walls to hydrolytic enzymes. This work demonstrates the potential of manipulating non-coding RNA to achieve both enhanced biomass and reduced cell wall recalcitrance, critical for the development of an environmentally friendly lignocellulosic biofuels industry.

## Results

### Overexpression of *miR408* enhances biomass yield

We generated overexpression (*miR408_OX*) and *miR408*-knockout poplars using a CRISPR/Cas9 genome editing approach (Supplementary Fig. 1a, b). After statistical analysis of plant height and stem diameter in three independent *miR408_OX* lines (#1, #5, and #6) (Supplementary Fig. 1c), two independent homozygous lines which had 218 bp genomic deletions (*miR408_cr* #8 and #20) were selected for further study (the identification of *miR408_cr* poplar is shown in Supplementary Fig. 2). The *miR408_OX* poplars displayed similar phenotypes of significantly increased plant height (34.75%, 20.42%, 16.94%) (Fig. 1a, b) and stem diameter (27.80%, 15.83%, 11.38%) (Fig. 1c), whereas the knockout lines showed no obvious changes in these parameters (Fig. 1a-c). In addition, overexpression or knockout of *miR408* did not result in a change of internode number compared with WT (Supplementary Fig. 1d). The net photosynthetic rate (NPR) of *miR408_OX* lines was nearly 40% higher than that of WT, whereas no significant difference between WT and knock-out lines was observed (Supplementary Fig. 1e).

qRT-PCR analysis revealed that mature *miR408* transcript was mainly present in mature leaves, young stems and roots (Supplementary Fig. 3a). In vascular tissue, *miR408* transcripts were highly expressed in vascular cambium and developing xylem, with the lowest expression in mature xylem (Supplementary Fig. 3a). To provide further insights into *miR408* expression pattern in the above organs, a 2-kb promoter region of *miR408* was fused with *GUS* gene and transformed into poplar. GUS signal was detected in leaf veins (Supplementary Fig. 3b, c) and in root vascular tissue (Supplementary Fig. 3d). Promoter activity of *miR408* was mainly detected in the vascular cambium that will differentiate into xylem, but was weak in mature xylem (Supplementary Fig. 3e-g).

Semi-thin sectioning showed that cells in the secondary xylem of *miR408_OX* plants were enlarged (Fig. 1d). The cambium zone of *miR408_OX* lines was wider by about 42% (Fig. 1d, Supplementary

Fig. 4a). Compared with WT, *miR408_OX* showed more xylem cells, xylem area (Supplementary Fig. 4b, c) and vessels (Supplementary Fig. 4d, e). Statistical analysis showed that vessel cell area in *miR408_OX* was significantly greater than in WT (Supplementary Fig. 4f, g), whereas there were no differences in fiber cell area (Supplementary Fig. 4h).

### Overexpression of *miR408* enhances saccharification efficiency

To test whether the enlarged xylem cells might possess more loosely-organized cell walls, we utilized green fluorescent protein (GFP)-tagged CBM1/3 (*Ct*CBM3-GFP and *Tr*CBM1-GFP) to identify exposed cellulose surfaces. The carbohydrate-binding module *Ct*CBM3 derived from the *Cladosporium thermocellum* cellulosomal scaffoldin protein (CipA) and *Tr*CBM1 derived from *T. reesei* cellobiohydrolase I specifically recognize the planar face of crystalline cellulose[25,26]. The area and intensity of the green fluorescence signal from *Tr*CBM1-GFP binding was much greater in tissue-cultured *miR408_OX* poplar than in the WT (Fig. 1e). Moreover, the green fluorescence signal appeared to cover a larger area of xylem tissues than in the *CCR2*-RNAi positive control (Fig. 1e, g), which has significantly enhanced saccharification[27]. However, the *miR408* knockout line #20 showed a very weak green fluorescence signal (Fig. 1e, g). Similarly increased cell wall accessibility of cellulose microfibrils to cellulase enzymes was seen for *Tr*CBM1-GFP (Supplementary Fig. 5a) and *Ct*CBM3-GFP (Supplementary Fig. 5b) binding to one-year-old natural dried poplar stems grown in the greenhouse. Notably, the labeling was highest in areas with reduced lignin (red autofluoresence) content.

Next, we examined the ability of green dye-labeled cellulase enzyme to bind to poplar stem cross sections. Unlike the CBM1/3-GFP labeling, all of the cell walls were labeled irrespective of the presence of lignin. However, the green fluorescence was very weak in the WT and much stronger in the *miR408_OX* plants, indicating strongly increased accessibility of cellulase to the cell walls of *miR408_OX* plants (Fig. 1f, h). Furthermore, in the cross sections of natural dried one-year-old poplar stems, the *miR408_OX* plants showed many more collapsed cells, probably due to water loss (Supplementary Fig. 6), whereas the cells were not collapsed in the cross sections of fresh material of the same lines (Supplementary Fig. 7).

Saccharification efficiency was determined by measuring the release of sugars from cell walls directly by digestion with cellulase, without acid pretreatment. The one-year-old natural dried poplar cell walls of the *miR408_OX* lines contained a slightly higher percentage of total sugars than the controls (Supplementary Fig. 8a). The total amount of enzyme-released sugar increased by 110-115% in the three *miR408_OX* lines (Supplementary Fig. 8b). When calculated as a percentage of total cell wall sugar, saccharification efficiency increased by 85-92% in the *miR408_OX* lines compared with the WT (Fig. 1i).

### Overexpression of *miR408* enhances biomass yield and accessibility to *Tr*CBM1-GFP of field-grown plants

After one-year growth in the field, the *miR408_OX* poplars displayed similar phenotypes of significantly increased plant height (42.57%, 41.06%, 38.20%) (Fig. 2a, b) and stem diameter (35.70%, 33.29%, 31.30%) (Fig. 2c) as seen with the greenhouse-grown poplars. *Tr*CBM1-GFP binding results showed the area and intensity of the green fluorescence signal was much stronger in the *miR408_OX* poplars (Fig. 2d), indicating the significantly increased cell wall accessibility of cellulose microfibrils to cellulase enzymes.

### *miR408* alters lignin deposition in poplar

Cross-sections with phloroglucinol staining showed that the basal stem xylem width of *miR408_OX* plants was significantly increased compared to WT (Fig. 3a, Supplementary Fig. 9a). No significant difference in secondary cell wall thickness of vessels and fibers was observed between *miR408_OX* and WT plants, but thickness was

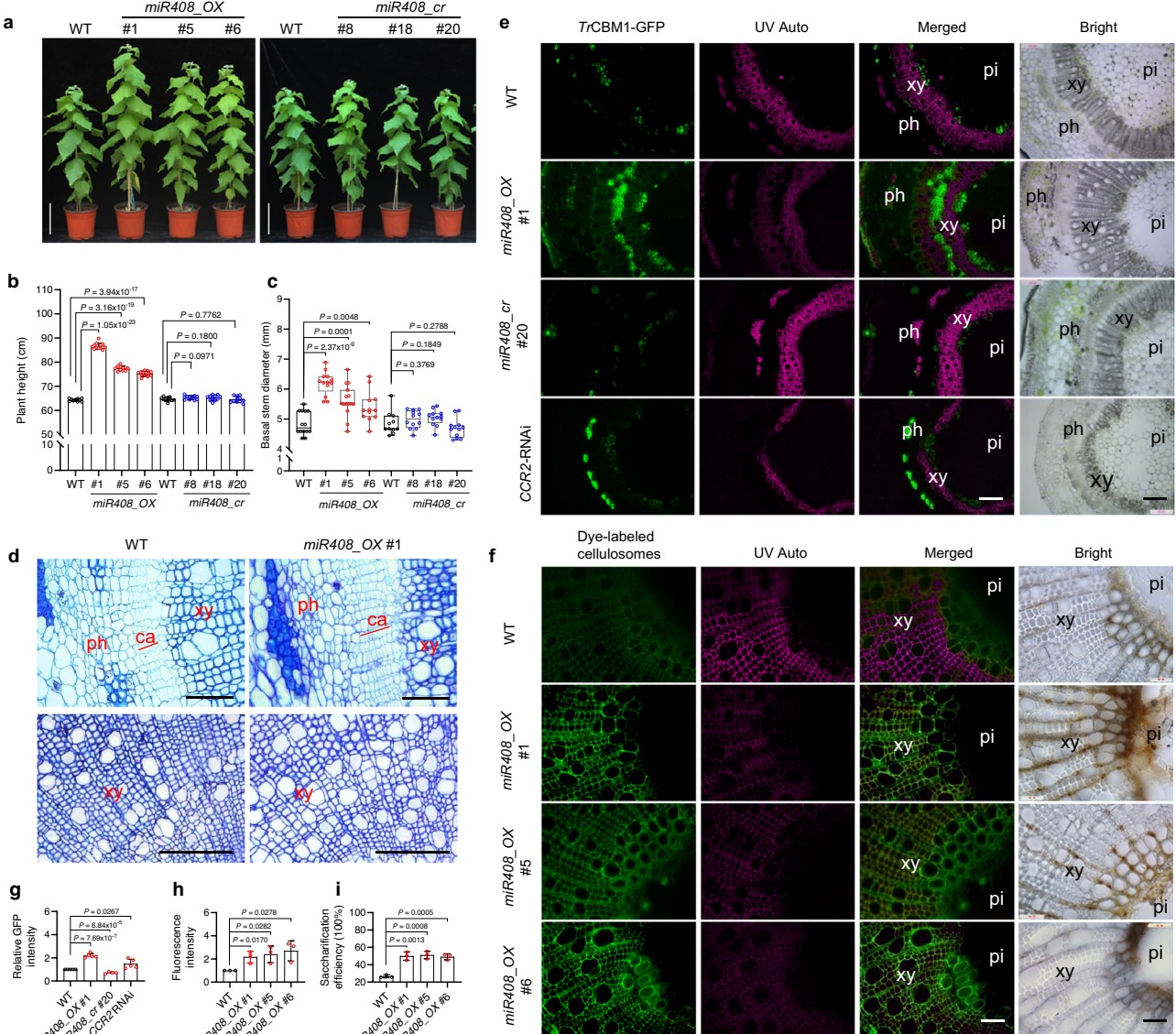

**Fig. 1 | Overexpression of *miR408* enhances biomass yield, cell wall accessibility and saccharification efficiency in poplar. a** Growth phenotypes of *miR408* overexpression and knockout lines. Bar, 15 cm. **b**, **c** Comparisons of plant height and basal stem diameter for the above lines. The upper and lower whiskers represent the maximum and minimum values, respectively. The upper, lower and middle box lines represent the two quartiles and median of values in each group. All *P*-values are from two-tailed Student's *t*-tests, n represents 11 to 21 trees sampled respectively from WT (*n* = 21), miR408_OX #1 (*n* = 13), #5 (*n* = 11), #6 (*n* = 11), miR408_cr #8 (*n* = 11), #18 (*n* = 11) and #20 (*n* = 11). **d** Comparison of vascular cambium zones and xylem area. Scale bar, 50 μm (upper); 100 μm (down). xy, xylem; ca, cambium; ph, phloem. Ten samples each were analyzed with similar results. **e** Fluorescence microscopy of cell walls exposed to *Tr*CBM1-GFP from WT, *miR408_OX* #1, *miR408_cr* #20 and *CCR2*-RNAi tissue cultured lines. The *CCR2*-RNAi line was used as the positive control. *Tr*CBM1 specifically recognizes cellulose and the probe exhibits green fluorescence. Autofluorescence (red) under UV shows lignin and the merged images highlight the negative correlation between probe binding and autofluorescence. Scale bars, 100 μm. **f** Fluorescence microscopy of transverse sections of one-year-old natural dried poplar stems from WT and *miR408_OX* exposed to dye-bound cellulases for 24 h. The overexpression lines showed much stronger green fluorescence than WT. Scale bars, 100 μm. Xy, xylem; pi, pith. **g**, **h** Histograms showing relative green fluorescence intensity of *Tr*CBM1-GFP (**g**) and dye-bound cellulase (**h**). **i** Saccharification efficiency of WT and *miR408_OX* plants without pretreatment. Values are presented as means ± SD (*n* = 3, All *P*-values are from two-tailed Student's *t*-tests; n represents 3 transgenic lines, and three replicate samples were carried out for each transgenic line). Source data are provided as a Source Data file.

decreased by around 50% in knockout poplars (Fig. 3b, Supplementary Fig. 9b). RNA-seq data showed that the expression levels of cell wall related NAC TFs such as *VND7* and *SND1* were down-regulated in *miR408_cr* poplar, and a negative cell wall related TF *LBD15* was highly up-regulated in *miR408_cr* poplar (Supplementary Fig. 9c). To determine whether the above changes in vascular morphology and saccharification efficiency were associated with changes in lignin deposition, we analyzed cross-sections from the first (IN1) to the eighth internode (IN8) with phloroglucinol staining (Supplementary Fig. 9d). Statistical analysis (Supplementary Fig. 9d, e) confirmed significantly

reduced number of lignified xylem cells, indicative of delayed secondary xylem differentiation, in *miR408_OX* compared with WT, with slightly increased number of xylem cells in knock-out poplars.

To understand the relationship between altered cell wall morphology and the changes in bulk lignin levels, we employed confocal Raman microspectroscopy (CRM) and stimulated Raman scattering (SRS). CRM and SRS techniques are applied to reveal the spatial distribution of intact plant cell wall components in their native form[28]. Stem cross-sections were analyzed using internode 20 (IN20), integrating over the 1600 cm⁻¹ band (1550⁻¹ to 1640 cm⁻¹) dominated by

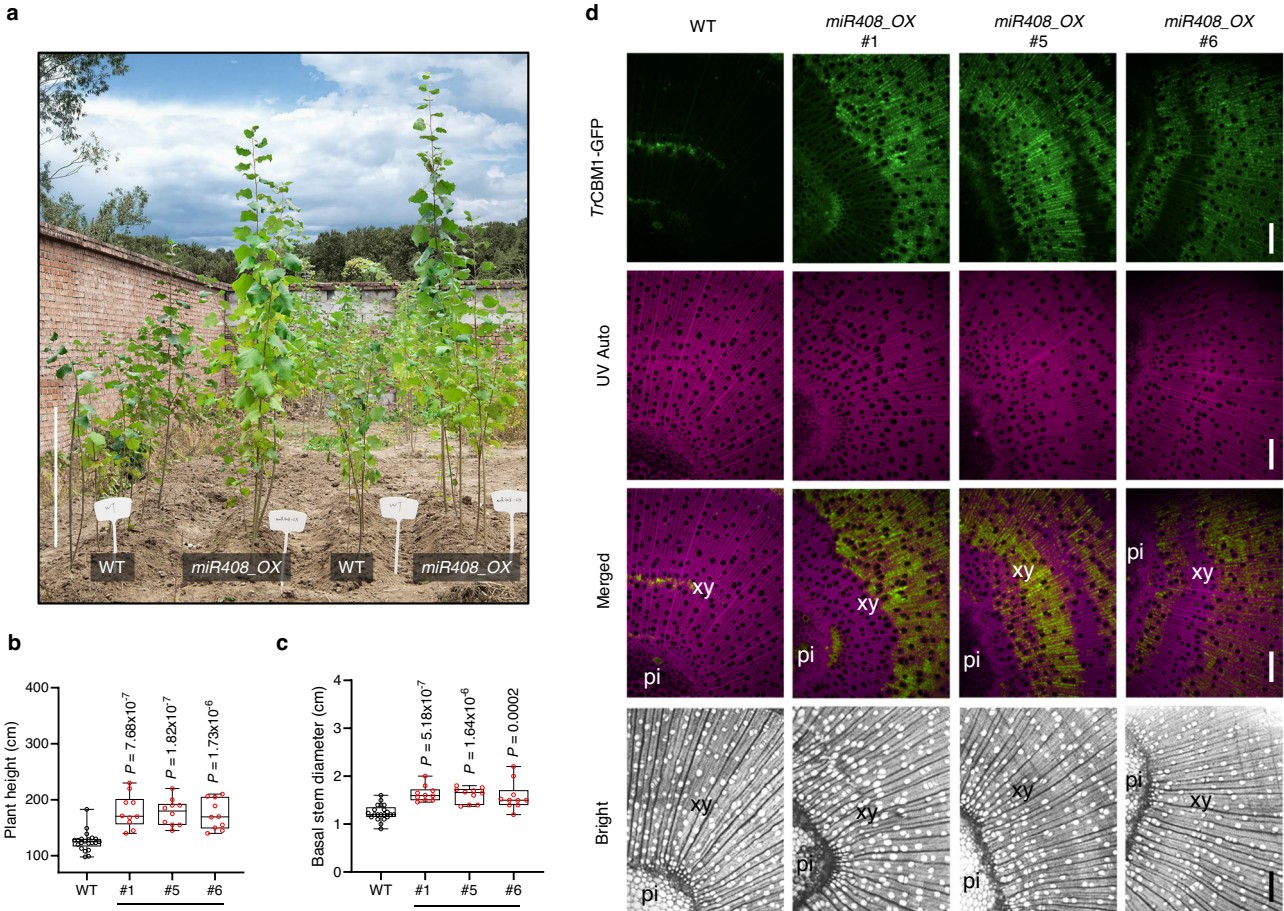

**Fig. 2 | Overexpression of *miR408* enhances biomass yield, cell wall accessibility and saccharification efficiency in field-grown plants. a** Growth phenotypes of *miR408* overexpression lines in the field. Bar, 90 cm. **b**, **c** Comparisons of plant height and basal stem diameter for the above lines. The upper and lower whiskers represent the maximum and minimum values, respectively. The upper, lower and middle box lines represent the two quartiles and median of values in each group. All *P*-values are from two-tailed Student's *t*-tests. n represents 10-21 trees sampled respectively from WT (*n* = 21) and *miR408_OX* #1, #5, #6 (*n* = 10). **d** Fluorescence microscopy of cell walls exposed to *Tr*CBM1-GFP from WT and *miR408_OX* poplars grown in the field. Bar, 100 μm. Six samples each were analyzed with similar results. Source data are provided as a Source Data file.

the aromatic C-C vibration (Fig. 3c). The lignin signal intensity in the secondary xylem cell walls in IN20 of *miR408_OX* plants was weak compared with that of WT plants (Fig. 3c), indicating a reduced lignin content. Furthermore, SRS with the quantitative analysis presented as pixel intensity showed that the SRS intensity of lignin across the cell wall was weaker in IN5-20 of *m*iR408 plants than in the corresponding internodes of WT plants (Supplementary Fig. 10).

Two-dimensional heteronuclear single quantum coherence nuclear magnetic resonance (2D-HSQC NMR) spectroscopy was performed on isolated lignin, using double enzymatic digestion. Only trace levels of H lignin units were detected. Lignin in *miR408_OX* lines had a lower relative abundance of S lignin units and a correspondingly higher abundance of G lignin units (Fig. 3d, f). The S/G ratio in three independent *miR408_OX* plants was 1.98, 2.02 and 2.21, compared with 2.88 in WT poplar (Fig. 3d). β-Aryl ether subunits (A) in *miR408_OX* poplars were relatively lower than in WT (Fig. 3e, g). Furthermore, the distribution of β−β (B) regions differed significantly between *miR408_OX* and WT plants (Supplementary Table 1). Because β-*O*−4 linkages are strongly correlated with S lignin, and a lower level of β-*O*−4 corresponds to a lower S/G ratio[29], these data are consistent with the compositional data revealed by correlations in the aromatic region.

To further quantify lignin and other cell wall polymers in *miR408_OX* poplars, woody stem tissues were analyzed using a range of wet chemistry approaches. Overall, the *miR408_OX* poplars showed a 10% decrease in lignin content compared with WT using the acetyl bromide (AcBr) method (Supplementary Fig. 11). The results obtained from Klason method revealed that the contents of both acid-insoluble lignin (AIL) and acid-soluble lignin (ASL) were decreased in *miR408_OX* stem compared with that in WT, with a decrease in total Klason lignin of around 4% (Table 1). Analysis of total cell wall carbohydrates indicated little change in cell wall glucose levels (reflecting cellulose), but reductions in xylose, mannose and glucuronic acid (reflecting hemicelluloses) in *miR408_OX* stem compared with that in WT (Table 1).

Analysis of acetylated lignin samples by gel permeation chromatography (GPC) method revealed reductions in the weight-average (Mw) and number-average (Mn) molecular weights, with increased poly-dispersity indexes (Mw/Mn), in *miR408_OX* compared to WT poplars (Table 2).

## miR408 targets LAC19, LAC25 and LAC32

To investigate the target(s) of *miR408* that influence cell wall structure and composition, RNA-seq analysis was performed on stems of overexpression lines (#1 and #6), CRISPR-Cas9 knock-out lines (*miR408_cr* #8 and #20) and WT poplars. We identified a total of 12,166 differentially expressed genes (DEGs) (Fig. 4a, Supplementary Fig. 12a). The down-regulated DEGs in *miR408_OX* poplars were enriched to the phenylpropanoid biosynthesis pathway (Fig. 4b, Supplementary Fig. 12b, c), indicating *miR408* may be involved in lignin biosynthesis.

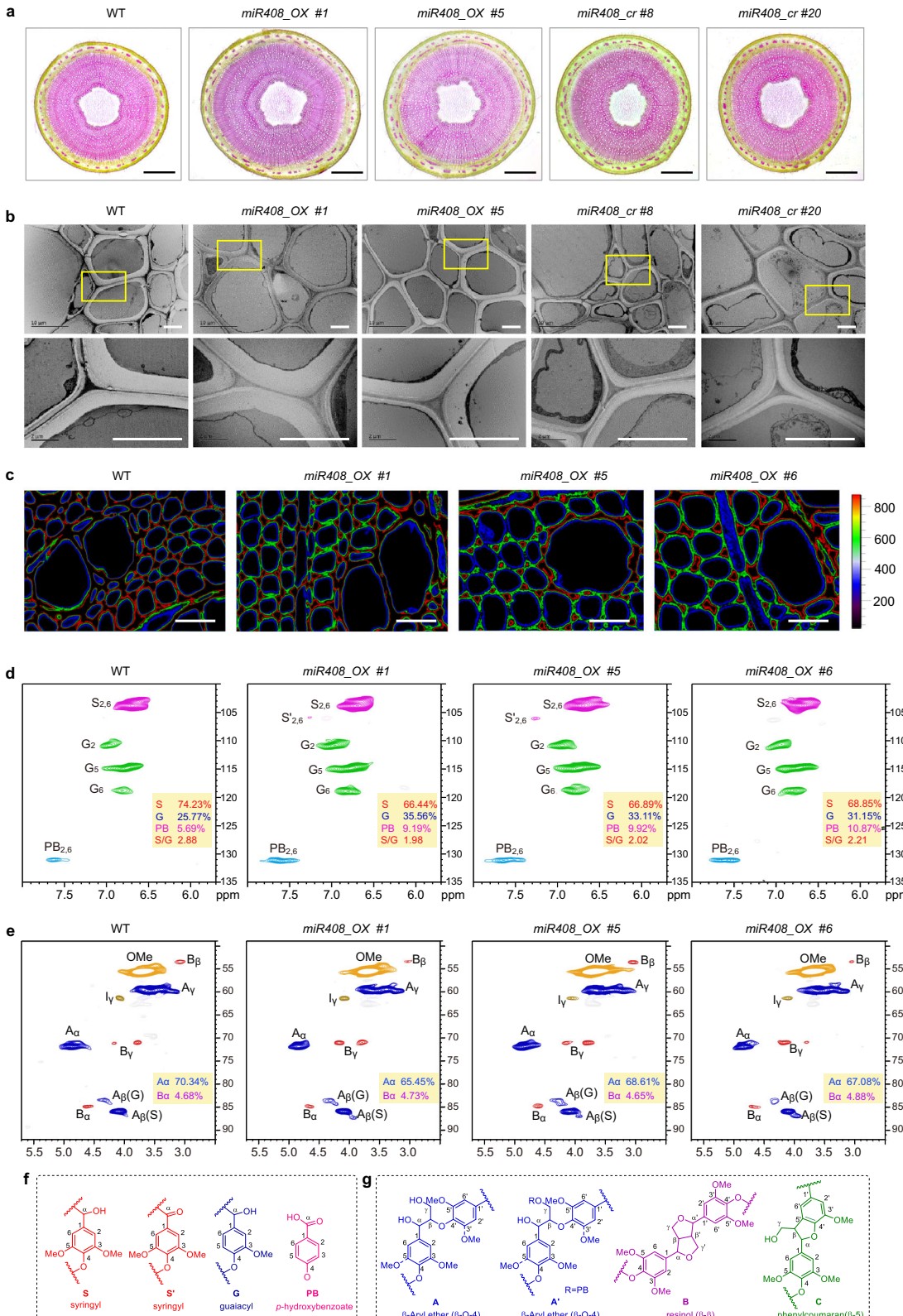

**Fig. 3 | *miR408* overexpression alters lignin deposition in poplar. a** Cross-sections of six-month-old basal stems of WT, overexpression and knockout poplar lines. Stem cross-sections were stained with phloroglucinol-HCl (red color indicates lignin). Scale bar, 1 mm. Ten samples each were analyzed with similar results. **b** Cell wall ultrastructure of mature xylem. Scale bar, 5 μm. Six samples each were analyzed with similar results. **c** CRM images of lignin in IN20 secondary xylem cell walls. The scale bar from *200–800* represents signal intensity from low to high. The red color represents a higher signal intensity than blue. Lignin is deposited throughout the cell wall in WT, whereas it is localized more in the corners of the cell walls of *miR408_OX* plants, indicating that *miR408_OX* plants have delayed lignification. Six samples each were analyzed with similar results. **d-g** 2D-HQSC NMR spectra revealing aromatic regions (**d, f**) and lignin aliphatic side chain regions (**e, g**). The lignin in *miR408_OX* poplar lines had a lower relative abundance of S lignin units and a correspondingly higher abundance of G lignin units. The β-aryl ether subunits (A) were lower in *miR408_OX* poplars.

**Table 1 | Cell wall composition of WT and *miR408_OX* poplars**

| | Lignin | | Total lignin | Cellulose | Hemicelluloses | | | Total Hemicelluloses |
|---|---|---|---|---|---|---|---|---|
| | AIL | ASL | | | Xyl | Man | GluA | |
| WT | 19.56 ± 0.03 | 5.03 ± 0.01 | 24.60 ± 0.04 | 39.16 ± 0.02 | 17.97 ± 0.04 | 0.95 ± 0.00 | 1.92 ± 0.01 | 20.84 ± 0.04 |
| *miR408_OX* #1 | 19.15 ± 0.03 | 4.54 ± 0.01 | 23.69 ± 0.04 | 39.26 ± 0.01 | 16.10 ± 0.01 | 0.60 ± 0.01 | 1.32 ± 0.01 | 18.02 ± 0.02 |
| *miR408_OX* #5 | 19.33 ± 0.02 | 4.28 ± 0.02 | 23.61 ± 0.04 | 39.59 ± 0.01 | 16.06 ± 0.01 | 0.61 ± 0.01 | 1.81 ± 0.01 | 18.48 ± 0.02 |
| *miR408_OX* #6 | 19.20 ± 0.02 | 4.32 ± 0.01 | 23.52 ± 0.03 | 40.61 ± 0.01 | 16.52 ± 0.01 | 0.62 ± 0.02 | 1.76 ± 0.01 | 18.90 ± 0.02 |

Contents of acid insoluble lignin (AIL), acid soluble lignin (ASL), total lignin, glucose (as an estimate of cellulose), and hemicellulosic monosaccharides. Values are means ± SE ($n$ = 3, where $n$ represents 3 transgenic lines, and three replicate samples were analyzed for each transgenic line). Values are expressed as weight percent based on vacuum-dried extractive-free wood weight (%, w/w). Source data are provided as a Source Data file.

**Table 2 | Weight-average molecular weight (Mw), number-average molecular weight (Mn), and poly-dispersity indices (PDI, Mw/Mn) of acetylated lignin samples from WT and *miR408_OX* poplars**

| | WT | *miR408_OX* #1 | *miR408_OX* #5 | *miR408_OX* #6 |
|---|---|---|---|---|
| Mw (g mol⁻¹) | 8697 ± 6.43 | 7313 ± 5.29 | 7309 ± 3.06 | 7186 ± 3.51 |
| Mn (g mol⁻¹) | 5033 ± 2.65 | 3967 ± 7.23 | 4021 ± 4.04 | 3941 ± 8.62 |
| PDI | 1.73 ± 0.01 | 1.84 ± 0.01 | 1.82 ± 0.01 | 1.82 ± 0.01 |

Values are means ± SE ($n$ = 3, where $n$ represents 3 transgenic lines, and three replicate samples were analyzed for each transgenic line). Source data are provided as a Source Data file.

In the *miR408_cr* poplar, the expression of key genes in lignin biosynthesis such as *COMT1* and *CCoAOMT* were nearly 2.3 times higher than that of WT plants, and the phenylpropanoid pathway genes, including *PAL1/2* and *C4H* were also up-regulated with fold change values 2.5 and 2.8, respectively (Supplementary Fig. 12c). These results indicated that lignin biosynthesis pathway is more active in *miR408_cr* plants, which is consistent with the increased intensity of phloroglucinol staining (Supplementary Fig. 9d). Using psRNAtarget prediction, three *LACCASES*, *LAC19*, *LAC25*, *LAC32* were predicted as the highest potential targets of *miR408* (Fig. 4c, Supplementary Data 1). qRT-PCR results showed *LAC19*, *LAC25* and *LAC32* transcripts were significantly decreased in *miR408_OX* plants while not obviously changed in *miR408_cr* poplars (Fig. 4d). This may be because the five predicated LACs, namely *LAC19*, *LAC25*, *LAC32*, *LAC47* and *LAC55* were also predicted to be targeted by at least one of the microRNAs such as *miR475*, *miR396*, *miR1447*, *miR397*, *miR169*, *miR7826*, *miR7466* and *miR7817* (Fig. 4c). There was no significant change in the transcript level of *LAC47* and *LAC55* in RNA-seq data (Supplementary Fig. 12d) or as determined by qRT-pCR (Supplementary Fig. 12e). Phylogenetic analysis indicated that Potri.013G152700 (*PtrLAC32*) is a homolog of *LAC13* of *A. thaliana*, a target of *miR408*[30]. Potri.008G073800 (*PtrLAC19*) and Potri.010G183500 (*PtrLAC25*) share the highest homology and are closely related to *LAC12* in *A. thaliana* (Supplementary Fig. 13).

Extractable laccase protein levels of *miR408_OX* plants were approximately 22.5 % lower on average than in WT (Supplementary Fig. 12f). *LAC19*, *LAC25* and *LAC32* were highly expressed in stems and root (Supplementary Fig. 12g-i), which are highly lignified tissues. In vitro assays using effector-reporter system of luciferase (Supplementary Fig. 14a, b) confirmed that *miR408* can negatively regulate the expression of *LAC19*, *LAC25* and *LAC32*. Based on 5´ RACE assay of *LAC19* (Fig. 4e), the issue 7/20 means that seven of twenty clones from the PCR products contained an *miR408*-guided cleavage 5´ end that mapped precisely to exon 2. Based on 5´ RACE assays of *LAC25* (Fig. 4f) and *LAC32* (Fig. 4g), the cleavage sites were all located at exon 2, and

six and seven, respectively from the twenty PCR products mapped precisely to the cleavage sites.

To verify the possibility of spatial interactions between *miR408* and *LACs*, fluorescence in situ hybridization (FISH) was carried out. Specific probes (FAM, CY5 and CY3) were designed to label *miR408*, *LAC19* and *LAC25* as green, pink and red, respectively. Lignin auto-fluorescence showed as the blue signal. In young stems, *miR408* was expressed in all the tissues, whereas *LAC25* was highly expressed in xylem and epidermal cells, and weakly expressed in phloem cells; *LAC19* was only expressed in the epidermal and parenchyma cells (Fig. 4h). In mature stems, *miR408* was expressed in all tissues, whereas *LAC25* was highly expressed in xylem, phloem fiber and cortical cells. *LAC19* was expressed in cortical and parenchyma cells, not in phloem fiber or xylem cells (Fig. 4i). The merged images of *miR408* and *LACs* showed there is a spatial correlation between *miR408* and *LAC19* and *LAC25*, and merged images of lignin and *LACs* are consistent with the involvement of the *LACs* in lignification.

## *lac19 lac25 lac32* poplar shows enhanced growth and saccharification efficiency

To explain the improvement of saccharification efficiency in *miR408_OX* poplar at the molecular level, we designed four sgRNAs to target *LAC19*, *LAC25* and *LAC32* (Supplementary Fig. 15a) to generate *lac19 lac25 lac32* loss-of-function plants. Meanwhile, we also constructed a *LAC19*, *LAC25* and *LAC32* vector to generate *LACs* overexpression poplar (Supplementary Fig. 15b). After PCR and sequencing, we obtained four independent homozygous lines of triple mutants (*lac19 lac25 lac32*, Supplementary Fig. 15c, d), five independent homozygous lines of double mutants (*lac25 lac32*, Supplementary Fig. 15e, f), and single mutant of *lac19* (Supplementary Fig. 15g, h; Supplementary Data 2). We analyzed the phenotypes of the mutants and overexpression poplars and found the triple mutants and double mutants showed both significantly increased plant height and diameter, while *lac19* showed slightly enhanced growth compare to the WT (Fig. 5a, e, f). The two-month-old tissue cultured *lac19 lac25 lac32* (Fig. 6a-c) also showed similar phenotypes to the soil-grown poplars. However, the three laccase overexpression poplars showed decreased plant height and basal stem diameter compared with WT (Fig. 5b, g, h).

To further address laccase function in cell wall lignification, we analyzed cross-sections of basal stems with phloroglucinol staining. Compared with the WT, *lac25 lac32* showed lighter lignin staining, and loose cell arrangement with a degree of vessel collapse (Fig. 5c). The cell walls of *lac19 lac25 lac32* showed more greatly decreased lignin staining, an even looser cell arrangement, and serious vessel collapse. The cell wall morphology of the single gene mutants of *lac19* was not as obvious as that of the triple and double mutants (Fig. 5c). The cell morphology of the double and triple *laccase* mutants was similar to that of the *miR408_OX* poplars. In contrast, *LAC19_OX*, *LAC25_OX* and *LAC32_OX* exhibited neatly arranged xylem cells, with similar morphology to WT (Fig. 5d). Compared with the WT, the three *LAC*

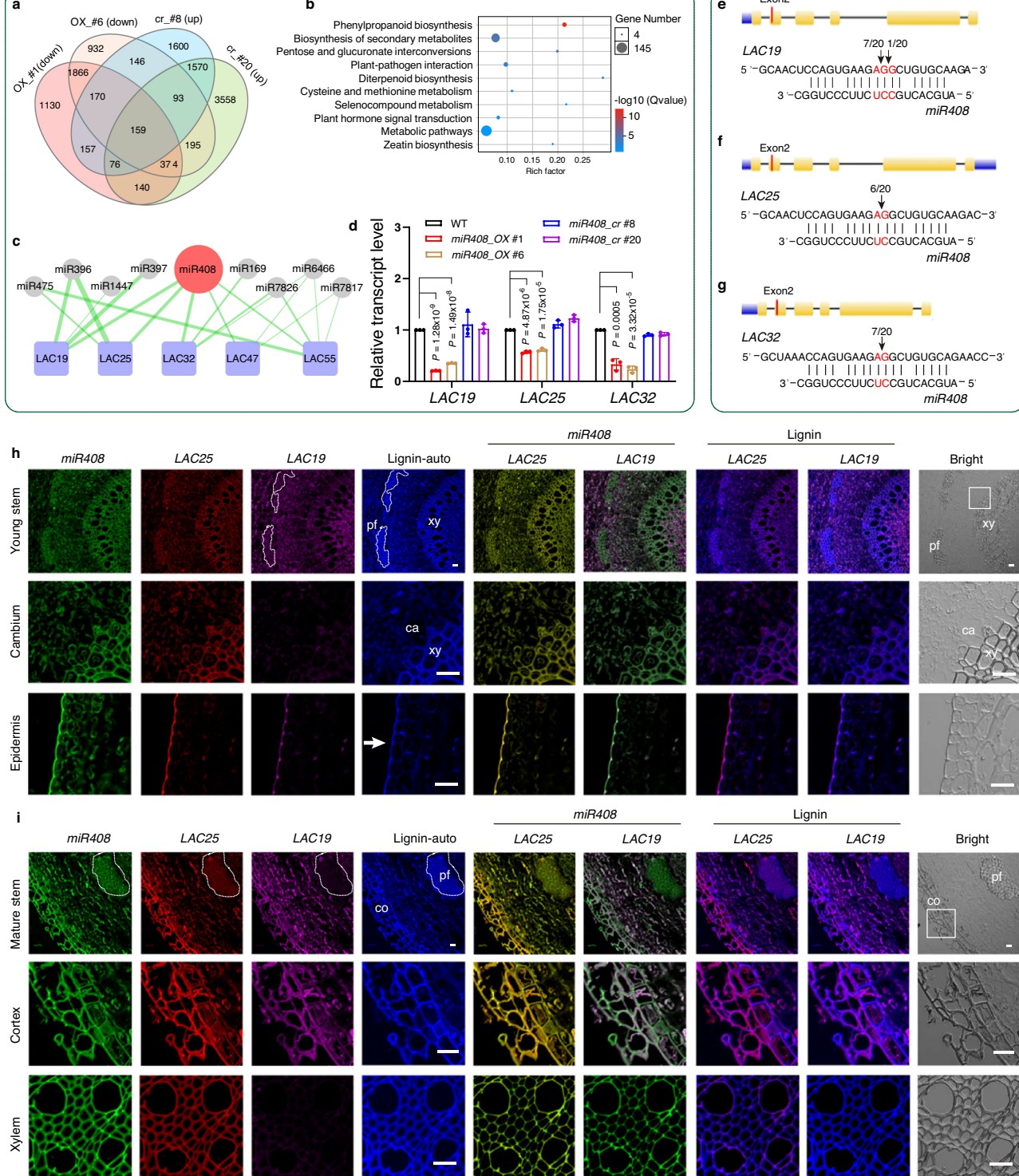

**Fig. 4 | *miR408* targets *LAC19, LAC25* and *LAC32*. a** Venn diagram comparison from RNA-Seq data analysis of *miR408_OX* and *miR408_cr* poplars. **b** KEGG pathway enrichment of DEGs that are down-regulated in *miR408_OX* poplars. **c** An interaction network of five predicated *LACs* (*LAC19, LAC25, LAC32, LAC47* and *LAC55*) with miR408 and other miRNAs. The thickness of the green line represents the strength of the interaction. **d** qRT-PCR showing the relative transcript levels of *LAC19, LAC25* and *LAC32* with high score using psRNAtarget in WT and *miR408_OX* and knockout plants. Values are means ± SD (All *P*-values are from two-tailed Student's *t*-tests., *n* represents 3 trees sampled respectively from *miR408_OX* #1, *miR408_OX* #6, *miR408_cr* #8 and *miR408_cr* #20). **e–g** 5′RACE assays showed the cleavage site of the *LAC19* (**e**), *LAC25* (**f**) and *LAC32* (**g**). The red lines in exon 2 show the target sites, and the black arrows show the detailed miR408-guided cleavage positions. **h, i** Localization of *LACs* and *Pag-miR408* using FISH method. Specific probes were designed to label *miR408*, *LAC19* and *LAC25* as green, pink and red, respectively. Lignin autofluorescence is displayed in blue. The merged images showed spatial correspondence between *miR408* and *LAC19* and *LAC25*. In young stem (**h**), *miR408* was expressed in all the tissues, while *LAC25* was highly expressed in xylem and epidermal cells, and slightly expressed in phloem cells; *LAC19* was only expressed in the epidermis and parenchyma cells. In mature stem (**i**), *miR408* was expressed in all the tissues, while *LAC25* was highly expressed in xylem, phloem fiber and cortical cells; *LAC19* was expressed in cortical and parenchyma cells, but not in phloem fiber or xylem cells. Scale bar, 20 μm. Three samples each were analyzed with similar results. The dashed box represents phloem fibers. Arrows indicate the epidermis. Boxes in h and i represent the enlarged area of cambium and cortex, respectively. Source data are provided as a Source Data file.

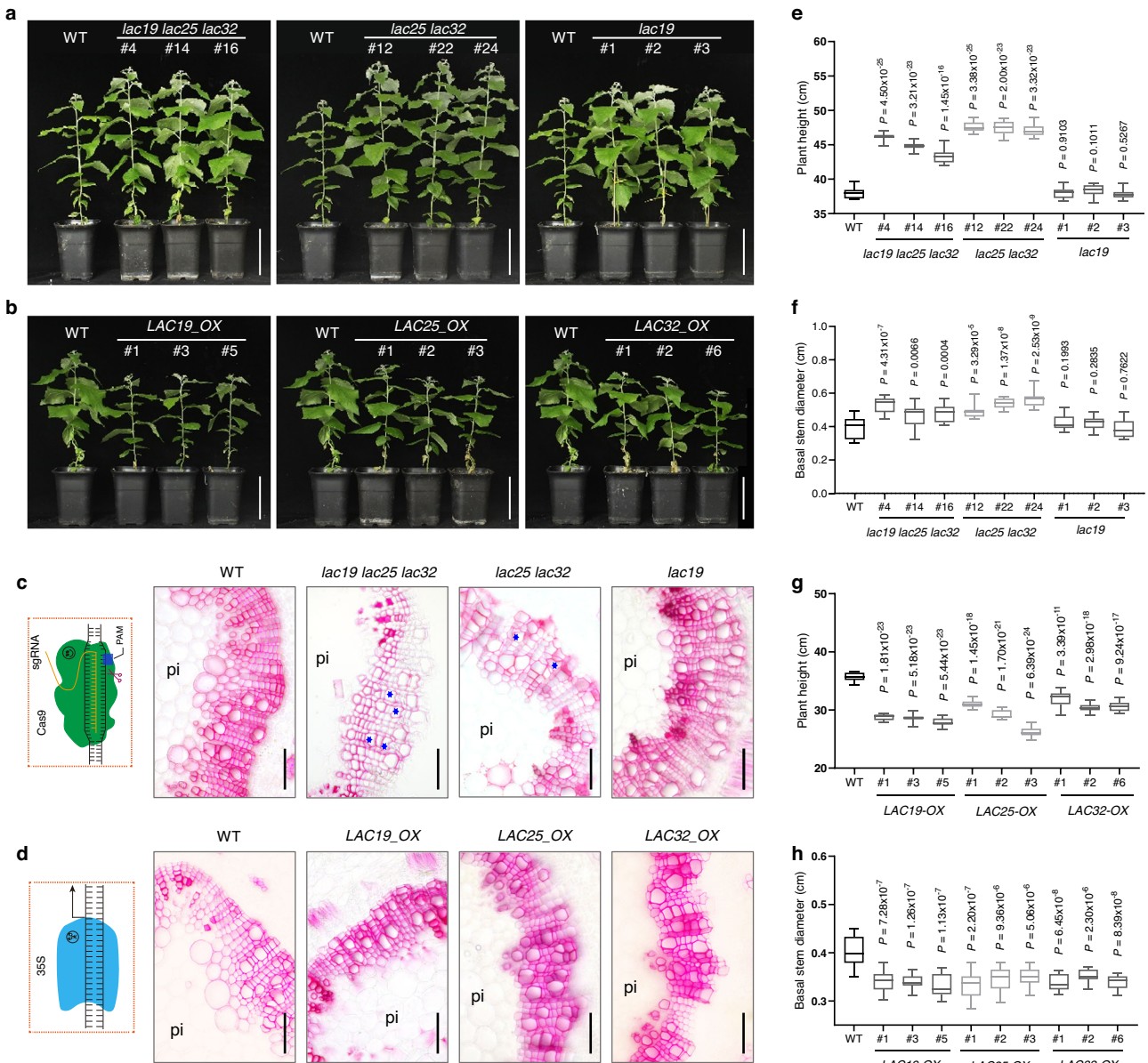

**Fig. 5 | *lac19 lac25 lac32* poplar shows enhanced growth. a, b** Growth phenotypes of *lac* mutants (**a**) and overexpression (**b**) poplars. Scale bar, 15 cm. (**c, d**) Phloroglucinol staining analysis of *lac* mutants (**c**) and overexpression (**d**) poplars. Scale bar, 40 µm. Compared with the WT, the cell walls of triple and double mutants showed lighter lignin staining and looser cell arrangement, with degrees of vessel collapse. Six samples each were analyzed with similar results. The images on the left represent CRISPR/Cas9-mediated-knockout (**c**) and 35S-mediated-overexpression

(**d**), respectively. The blue asterisks represent collapsed vessels. **e–h** Comparisons of plant height (**e, g**) and basal stem diameter (**f, h**) for the *lac* mutants (**a**) and overexpression (**b**) poplars. The upper and lower whiskers represent the maximum and minimum values, respectively. The upper, lower and middle box lines represent the two quartiles and median of values in each group. All *P*-values are from two-tailed Student's *t*-tests, and *n* represents 15 trees sampled respectively from each line in **e, f, g, h**. Source data are provided as a Source Data file.

overexpression transgenic lines showed stronger staining (Fig. 5d). Klason analysis showed lignin content of the *LAC19_OX*, *LAC25_OX* and *LAC32_OX* plants were increased by 19.69%, 10.67% and 14.62% respectively (Supplementary Table 2).

To explore whether laccases are the key target for enhancing saccharification efficiency of *miR408_OX* poplars, cell wall accessibility of cellulose microfibrils to cellulase enzymes of the *lac* mutants was determined, using untreated basal stem cell walls of two-month-old tissue cultured poplars. In *Tr*CBM1-GFP binding experiments, the *lac19 lac25 lac32* lines showed significantly increased green fluorescence signal compared with WT in both phloem fiber and xylem, the *lac19 lac25* lines also showed increased green fluorescence signal, whereas *lac19* poplar was similar to WT (Fig. 6d). From the statistical analysis of fluorescence intensity, the green fluorescence signal of phloem fibers

and xylem was highest in *lac19 lac25 lac32* poplar (Fig. 6e). Furthermore, *lac* mutants showed delayed phloem fiber development compared with WT (Fig. 6d).

In green dye-labeled cellulase enzyme binding studies, the green signal was just developing in WT (Supplementary Fig. 16a-c) after 30 min incubation, whereas it was clearly seen in the cell walls of *lac19 lac25 lac32* stems by this time (Supplementary Fig. 16d, e). In contrast, labeling of cortical cells was much reduced in *lac19 lac25 lac32* (Supplementary Fig. 16c, f). The *lac25 lac32* (Supplementary Fig. 16g-i) lines also showed increased cell wall accessibility of cellulose microfibrils to cellulase enzymes compared with WT (Supplementary Fig. 16a-c), whereas *lac19* did not (Supplementary Fig. 16j, k). The results of saccharification assays were consistent with the microscopic results (Supplementary Table 3). Next, we obtained CRM images (Fig. 6f) and

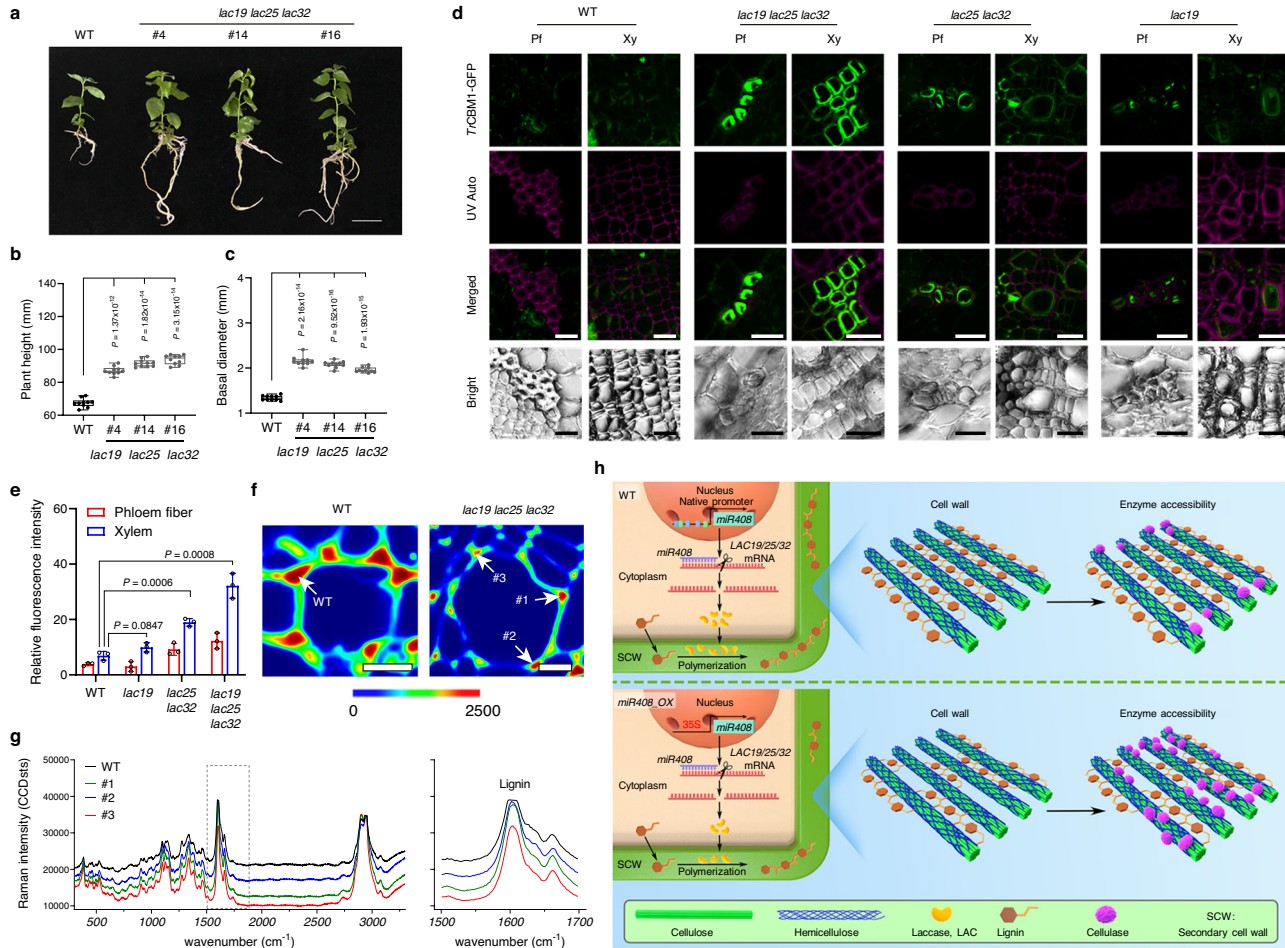

**Fig. 6 | The *lac19 lac25 lc32* triple mutant shows enhanced saccharification efficiency. a** Growth phenotypes of *lac19 lac25 lc32* triple mutant. Bar, 40 mm. **b**, **c** Statistics analysis of plant height (**b**) and basal diameter (**c**) of the poplar plants (**a**). The upper and lower whiskers represent the maximum and minimum values, respectively. The upper, lower and middlebox lines represent the two quartiles and median of values in each group. All *P*-values are from two-tailed Student's *t*-tests, and n represents 10 trees sampled respectively from each line. **d** Fluorescence microscopy of cell walls of WT, *lac19 lac25 lac32*, *lac25 lac32*, *lac19* lines exposed to *Tr*CBM1-GFP. *Tr*CBM1 specifically recognizes cellulose and the probe exhibits green fluorescence. Autofluorescence (red) under UV shows lignin and the merged images highlight the negative correlation between probe binding and auto-fluorescence. Pf, phloem fiber; xy, xylem. Scale bars, 20 μm. **e** Histograms showing relative green fluorescence intensity of *Tr*CBM1-GFP. Values are means ± SD (*n* = 3). All *P*-values are from two-tailed Student's *t*-tests. *n* represents 3 trees sampled respectively from WT and the mutants). **f** CRM images of *lac19 lac25 lac32* vessels, showing lignin deposition. Scale bars, 10 μm. Arrows indicate cell corners of WT and *lac19 lac25 lac32* tested in **g**. **g** Spectra analysis of vessel cell corners in **f**. The grey dotted-line box represents lignin Raman spectra and is enlarged in the right panel of **g**. **h** A proposed model for the role of *miR408* in regulating poplar lignification and saccharification efficiency. Monolignols are synthesized in the cytoplasm and then exported to the apoplast, oxidized and integrated into the secondary cell wall. This polymerization process is initiated by laccases, which can be down-regulated by *miR408*. When *LAC19, LAC25* and *LAC32* were down-regulated, the polymerization of monolignols is repressed, resulting in decreased lignin content and molecular weight, leading to less polymer–polymer cross-linking of lignin and cellulose, and exposing more cellulose scaffold, with increased wall porosity and reduced wall cohesiveness, which allows increased enzyme access to biomass during saccharification in *miR408_OX* poplar. Source data are provided as a Source Data file.

spectra (Fig. 6g) to investigate changes in lignin content of vessels in *lac19 lac25 lac32* and WT. The lignin distribution images were acquired by integral of the *1,545* to *1,710* cm⁻¹ region originating from symmetrical stretching of aromatic rings. Clearly, the CRM images showed that the concentrations of lignin were highest in cell corners, and *lac19 lac25 lac32* showed significantly decreased lignin deposition compared with WT (Fig. 6f, g).

## Discussion

Previous studies showed that over-expression *miR408* can promote vegetative growth, while the impaired growth was observed in *miR408* T-DNA insertion *Arabidopsis* mutant lines[30]. In addition, cell wall thickening was associated with the deposition of lignin and cellulose[31,32]. In our study, the knock-out of *miR408* in poplar resulted in the enhanced cell wall lignification but the reduced cell wall thickness. Although lignin-related genes were up-regulated in the *miR408* knock-out plants from the RNA-seq data, the genes encoding activators (VND7 and SND1)[33–35] in the transcriptional regulatory network pathway of secondary wall synthesis were down-regulated. Moreover, the expression level of *LBD15*, a key TF that can down-regulate the expression of cellulose synthesis genes[36], was increased to a large extent in *miR408* knock-out poplars. The genetic evidence and gene expression analysis together suggested that *miR408* may possess an additional role of regulating cell wall thickening in plants.

Overexpressing *miR408* in poplar contributed the enhanced plant growth associated with a significant increase in net photosynthetic rate, somewhat similar to the phenotype when overexpressing *miR408* in *Arabidopsis*[24]. In addition, we found *miR408_OX* poplar showed wider cambium zone and increased xylem area (with enlarged xylem cells). We also showed that *miR408* was mainly expressed in the

vascular cambium and developing xylem. These results suggest a specific function for *miR408* in wood formation, which is characterized by sequential differentiation of vascular cambial cells into xylem cells, cell expansion, massive deposition of secondary cell walls, programmed cell death, and ultimately, the formation of heartwood[37,38]. Although these anatomical features were clearly linked to *miR408* overexpression or knock-out, more work is necessary to understand the coordination of the developmental and biochemical changes.

It was worth noting that overexpression of *miR408* in poplar, for both laboratory- and field-grown plants, results in a large increase in saccharification efficiency with no requirement for acid-pretreatment. Our study found that overexpression of *Pag-miR408* can target *Pag-LAC19,25,*32, delay lignification, and modestly reduce lignin content, S/G ratio and degree of lignin polymerization. The *lac* triple mutants showed similar phenotypes in the vascular cell morphology, cell wall accessibility, and saccharification with *miR408_OX* poplars. Laccases are considered to function in the polymerization of lignin monomers, potentially at the stage of polymer initiation, and subsequently in concert with peroxidases[39]. Given that reduced degree of lignin polymerization is associated with improved lignin extractability and reduced biomass recalcitrance[40], we speculate that the changes in lignin distribution and composition observed in the present *miR408_OX* plants result largely from the post-transcriptional regulation of three target *LACCASES*. Both decreased lignin polymer size and delayed lignification in *miR408_OX* indicated a mechanistic basis for the improved saccharification efficiency, which is enhanced by the altered ultrastructure of the vascular tissues, brought about by delayed secondary cell wall deposition during development. As shown by the increased accessibility of vascular tissues to *C. thermocellum* CBMs and fungal cellulase, we can conclude that the high saccharification efficiency of *miR408_OX* plants was largely linked to the more open cell walls for (downstream) deconstruction process.

Carbohydrates found in lignocellulose (cellulose and hemicellulose) are an abundant and unused source of raw material for producing commodity chemicals such as ethanol. Depending on the gene target and extent of down-regulation, targeting monolignol biosynthetic genes can improve saccharification efficiency with or without negative impacts on biomass[6,10,41–43], but seldom if ever with positive impacts. Lignification and growth are opposing processes competing for cellular resources. Here we demonstrate that it is possible to engineer directionally opposite changes in these two processes with a single transgene. In the model in Fig. 6h, we suggest that the decreased recalcitrance phenotypes and enhanced growth of the *miR408_OX* biomass result from repressed polymerization of monolignols, leading to less polymer–polymer cross-linking of lignin, increased wall porosity and reduced wall cohesiveness, which together increase enzyme access to biomass during saccharification. On the other hand, the looser microfibril structure may be easier to expand when the cell is growing under turgor pressure.

In conclusion, down-regulating multiple laccases through targeting of specific miRNAs may be a promising way to enhance biomass saccharification through reducing lignin polymerization. These findings, which are translatable to the field, can facilitate generation of improved tree feedstocks coupling enhanced saccharification efficiency with high total biomass, providing a promising and effective approach to the production of lignocellulosic bioenergy.

## Methods

### Plant materials and growth conditions
The hybrid poplar '84 K' (*Populus alba* × *P. glandulosa*) was used for genetic transformation. After transgenic plants were generated, screened, and verified, they were grown in a phytotron for 3, 6 and 12 months with a photoperiod of 16 h/8 h (light/dark) at 22 °C before detailed characterization. The *miR408_OX* and WT poplars were grown in a field plot on Beijing Forestry University field garden (40.01° N,116.35° E). After one-year growth, the plant height and basal stem diameter were measured and poplars were harvested for saccharification efficiency analysis.

### Plasmid construction and plant transformation
Genomic sequences containing *Pag*-pre-*miR408* on chromosome 2 were amplified via PCR from genomic DNA isolated from 84 K. The fidelity of the *Pag*-pre-*miR408*, the CDS of *LAC19*, *LAC25* and *LAC32* amplification was confirmed by sequencing and it was inserted into pCAMBIA2300 using *Bam*H I and *Kpn* I (Clontech) driven by the CaMV35S promoter. The 2000 bp *Pag-miR408* promoter region was amplified by PCR using the primers *Pag-miR408* pro-F and *Pag-miR408* pro-R to create *Pag-miR408* pro::*GUS*, which was inserted into pBI101 using *Xma* I and *Sal* I (Clontech), and confirmed by sequencing. To generate CRISPR-Cas9 *miR408* knock-out poplars (*miR408_cr*), four sgRNAs were designed, two upstream and two downstream, using CCTop-CRISPR/Cas9 target online predictor (https://cctop.cos.uni-heidelberg.de:8043/). The *miR408*-knockout vector harboring a GUS marker gene for easily identification of positive poplars. The DNA fragments encoding the selected sgRNAs1, 2, 3 and 4 were as follows: 5′-GAGAAGGCTGAGGCTTTGAGAGG-3′, 5′-GGAAGCACAATGAAAGGT GAAGG-3′, 5′-CCTCCCTTCTTTTGTTTCCATTA-3′, 5′-GAGCGAAATATA ACAGCAGCAGG-3′. To knock out *LAC19*, *LAC25* and *LAC32* (*LACs_cr*) at the same time, four sgRNAs1, 2, 3, 4 were designed: 5′-CGTCT CTCTTTGTTCTTCTTGGG-3′, 5′- GTCTTGCACAGCCTCTTCACTGG −3′, 5′- CTGCTTCAGAGAAATGAGCTTGG −3′, 5′- GTTCTGCACAGCCTC TTCACTGG −3′. The sgRNA2 can knock out *LAC19* and *LAC25* at the same time. The *miR408_OX*, *miR408_cr*, *LACs_OX* and *LACs_cr* and *pmiR408::GUS* fusion vectors were introduced into *Agrobacterium tumefaciens* GV3101 for transformation. To provide a positive control, we generated poplar lines with lack of expression of the *cinnamoyl CoA reductase CCR2* gene through RNAi. To generate the *PtCCR2*-RNAi line, the amplified fragment of *Pag-CCR2* was cloned into PK7GWIWG2(II) RR vector by Gateway LR reaction. The two-month-old *CCR2*-RNAi tissue cultured line showed nearly 40% reduced plant height and 50% reduced stem diameter. All the primers shown in Supplementary Data 3 were synthesized by BGI. White Co., Ltd. All the constructs were transformed into '84 K' by the *A. tumefaciens*-mediated leaf disc method according to a previous protocol[44]. For the sequence of all the mutants, the genome sequence including the targets were amplified and inserted into pGM-T. Primers are shown in Supplementary Data 3.

### RNA isolation and quantitative real-time PCR
To determine the expression pattern of *miR408*, young leaves (first to third from the top), mature leaves (fifteenth from the top), young stems (first to third from the top), and tissues of the phloem, vascular cambium, developing secondary xylem and mature xylem from internodes 11–14 of six-month-old vigorously growing '84 K' poplars were harvested and stored in liquid nitrogen until use. Total RNA, including small RNAs, was isolated using an RNA isolation kit (TIANGEN, DP501). Reverse transcription was performed using a First-Strand cDNA Synthesis Kit (TIANGEN, KR211). Specific primers were designed for analysis of *Pag-miR408* transcript levels (Supplementary Data 3). 5 S rRNA was used as the endogenous control to normalize the relative expression levels of *miR408*. qRT-PCR was performed using SYBR Green Mix (TIANGEN, FP401). For miRNA targets, the forward and reverse primers were designed upstream and downstream of the cleavage site. 18 S RNA from *P. trichocarpa* was used as an endogenous reference. In each reaction, 0.3 mM primer and 10 ng cDNA were used. For each of three biological replicates, PCR was performed in triplicate. The initial denaturing time was 30 s, followed by 40 cycles of 95 °C for 15 s, 60 °C for 15 s, and 72 °C for 15 s, with a final extension at 72 °C for 10 min. A melting curve was performed after the PCR cycles. All primer sequences are shown in Supplementary Data 3.

## RNA-seq analysis

RNA-seq analysis was performed with total RNA isolated from differentiating stem xylem of six-month-old poplars. Stems from three different poplar plants were pooled as one biological replicate. The transcriptome data were analyzed according to a previously described method[45]. Briefly, RNA-Seq data were generated with an Illumina HiSeq 2000 instrument at Novogene Ltd. (Beijing, China). The cleaned reads were mapped to the *P. trichocarpa* genome v3.1 (https://phytozome-next.jgi.doe.gov/info/Ptrichocarpa_v3_1) using the Hisat2 algorithm. The gene expression levels were calculated using Fragments Per Kilobase of exon model per Million mapped fragments (FPKM) method. Cytoscape v3.7.1 software was used for *miRNAs-LACs* network visualization (https://cytoscape.org/). The heatmaps were generated by using the GraphPad Prism v8.3 software (https://www.graphpad.com).

## Determination of saccharification efficiency

Cell wall residues (CWR) generated for lignin analysis were also used to analyze the amount of total sugar. Cell wall residue refers to the cell wall material extracted by methanol and chloroform after grinding through a 40-mesh sieve. The specific extraction process is as follows: cell wall residues were generated by extracting plant tissue with methanol (three times at 37 °C for 1.5 h) and chloroform: methanol (2:1) (three times at 37 °C for 1.5 h). The samples were then washed three times with water at 37 °C for 1.5 h and lyophilized for 48 h[41]. Enzymatic saccharification of stem samples was carried out for one-year-old *miR408_OX* grown in the greenhouse and ten-month-old *miR408_OX* poplars grown in the field. For total sugar and sugar component assay, $100 \pm 0.2$ mg CWR was weighed into a 5 mL glass vial, 1.5 mL 72% (w/w) $H_2SO_4$ added, and the mixture incubated in a water bath at 30 °C for 1 h, shaking every 10 min. Samples were transferred into 50 mL glass bottles with 42.0 mL MilliQ $H_2O$, and autoclaved at 121 °C for 1 h. Materials were transferred into new 50 mL tubes, 5-10 mL Milli Q $H_2O$ was added, and the tubes centrifuged at $2,600 \times g$ for 30 min. The supernatants were transferred into new 50 mL tubes for sugar assay. For measuring enzyme-released sugar and sugar components without acid pretreatment, $100 \pm 0.2$ mg CWRs were weighed into 15 mL tubes, 10 mL Enzyme Digestion Solution added, and the tubes placed in an angled shaker and shaken at 100 rpm for 72 h at 50 °C. The tubes were centrifuged and the supernatants used for assay of released sugars[10]. The amount of fermentable sugars was analyzed using the phenol-sulfuric acid assay method[46]. Saccharification efficiency was determined as the ratio of sugars released by enzymatic hydrolysis for 72 h to the amounts of sugars present in the cell wall material before enzymatic hydrolysis.

## Determination of lignin content

Debarked increment cores from one-year-old *miR408_OX* grown in the greenhouse were dried at 40 ºC. After drying, the wood chips were ground in a Wiley mill with a 40-mesh screen. The resulting wood meals (40–60 mesh) were used for determination of AcBr lignin content[47]. Neutral sugars in the acid-soluble fraction were derivatized to alditol acetates for quantitation by gas chromatography using a flame ionization detector (GC-FID; Agilent 7890 A)[48]. The AcBr method was used to quantify lignin content[49].

## Determination of structural carbohydrates and lignin in biomass

One-year-old natural debarked dried *miR408_OX* poplar stems grown in the greenhouse were collected. The chemical compositions (cellulosic and hemicellulosic monosaccharides and lignin) of raw material and pretreated substrates were analyzed by the standard procedure of the National Renewable Energy Laboratory (NREL)[50]. Briefly, ground (40-60 mesh) stem material (0.3 g) was placed in a tared pressure tube, 3 mL of 72% $H_2SO_4$ (w/w) added, and the pressure tube placed in a water bath at 30 °C and incubated for I h. The acid was diluted to a 4% concentration by adding 84 mL deionized water using an automatic burette prior to further analysis. For carbohydrate determination, sugar recovery standards (SRS) were made by weighing out 0.1 mg of each sugar, adding 10 mL deionized water and 348 μL of 72% sulfuric acid, and transferring to a pressure tube and capping tightly. For the lignin determination, acid soluble lignin and acid insoluble lignin were determined separately[50].

## Characterization of growth phenotypes

For the *miR408_OX* and *miR408_cr* poplars, the growth phenotype was observed and photographed after six months of growth in the phytotron. The *LACCASE* overexpression and knockout poplars were photographed after three months. To investigate the anatomical structures of stems, basal stems were fixed with 2.5% (w/v) glutaraldehyde in 0.1 M cacodylate buffer (pH 7.2) for 24 h at 4 °C. The samples were then rinsed in 0.1 M cacodylate buffer, dehydrated in a graded ethanol and acetone series, infiltrated, and embedded in Spurr resin (Electron Microscopy Sciences). The resin blocks were polymerized for 24 h at 70 °C. A microtome (Leica-RM2265, Germany) was used to produce 1μm-thick sections. The sections were stained with 1% (w/v) toluidine blue and imaged using a digital camera equipped with a Leica Aperio VERSA digital pathology scanner (Leica, Germany). To analyze cell wall ultrastructure, mature xylem samples of basal stems were fixed with 2.5% (w/v) glutaraldehyde in 0.1 M cacodylate buffer (pH 7.2) for 24 h at 4 °C, rinsed in 0.1 M cacodylate buffer, and samples were fixed with 1% (w/v) osmic acid for 12 h at 4 °C. After the same embedding process, 40-nm slices were cut using an ultramicrotome and were visualized directly by transmission electron microscopy (TEM).

## Image-Pro Plus analysis

Image-Pro Plus 6.0 (Media Cybernetics) software was used to quantify cell wall thickness, secondary xylem width, cell number and cell area[51].

## GUS staining and analysis

Stem sections were hand-cut and then incubated in 90 % acetone (v/v) for 10 min on ice then in GUS staining solution (100 mM sodium phosphate (pH 7.0), 10 mM EDTA, 0.5 mM ferricyanide, 0.5 mM ferrocyanide, 0.1% Triton X-100, 20 % (v/v) methanol, and 2 mM X-GLUC at 37 °C for 3 h. Following staining, sections were cleared in 95% ethanol, preserved in 75% ethanol and then photographed under the microscope (Leica DM2500, Germany).

## Wiesner staining

Cross-sections of *miR408_OX* poplars were cut from stems collected after 6 months of growth to examine the increase in stem diameter using a sliding microtome (Leica SM2010 R, Germany). Stems were also collected at 3 months for tracking the development of lignification using hand-sectioning for the very young internodes and the sliding microtome for older internodes. Phloroglucinol (0.3 g, MACKLIN) was dissolved in 10 mL absolute ethanol to prepare a 3% phloroglucinol solution[52]. Fresh stem sections were placed on a slide, and 50 μL phloroglucinol solution dropped onto the section, followed by 50 μL concentrated hydrochloric acid. A cover slip was added, color development observed, and the staining photographed under the microscope (Leica DM2500, Germany).

## Phylogenetic analysis

The amino acid sequences of 17 laccases in *Arabidopsis* were downloaded from the *Arabidopsis* information resource (https://www.arabidopsis.org/). The amino acid sequences of 56 laccases in poplar were blasted from *P. trichocarpa* v3.1 (https://phytozome-next.jgi.doe.gov/info/Ptrichocarpa_v3_1). The amino acid sequences of full-length laccases were aligned and analyzed with Clustal X 2.1 software. OrthoFinder v2.3.1 and MEGA-X 7 software were used for phylogenetic

analysis[53], with the neighbor-joining algorithm and bootstrap method of 1,000 replications. The aligned amino acid sequences used for phylogenetic analysis in Supplementary Fig. 13 are presented in Supplementary Data 4.

## Assay of laccase protein levels

Equal samples of fresh poplar stems of not less than 50 mg were accurately weighed. A corresponding volume (10% tissue homogenate ratio, w/v) of extraction buffer (10 mM PBS, pH 7.2–7.4) was then added and the stem samples were thoroughly homogenized. The homogenate was centrifuged twice at 4 °C for 10 min each, at 13,000×$g$ and the sediment discarded. The supernatant was centrifuged at 4 °C for 15 min at 13,000×$g$. The supernatant represents the isolated stem protein extract. A double antibody sandwich ELISA assay was used to detect laccase protein level according to the manufacturer's instructions (MEIMIAN).

## Raman analysis

Raman spectra and Raman mapping were acquired/performed with a Via-Reflex 532/XYZ confocal Raman system (Renishaw, Great Britain). Fresh samples of 3-month-old WT and *miR408_OX* plants were cut into approximately 10-μm-thick sections using a sliding microtome (Leica SM2010 R). The cross-sections were focused with 532 nm laser beams and the 50× objective was used for spectroscopic analysis. The laser power on the sample was approximately 5 mW. The Raman light was detected by an air-cooled spectroscopic CCD, and the output wavelengths of the optical parametric oscillators were selected at the *1,600* cm$^{-1}$ lignin aromatic ring vibration. An integration time of 1 s and a mapping step of 0.5 mm for a 10 × 10 mm region were chosen and each pixel corresponded to one scan.

## SRS microscopy

Cross-sections of 50 μm thickness were cut at internodes IN5, IN9, IN15 and IN20 from *miR408_OX* poplars, using a razor blade. The SRS imaging microscope using a mode-locked Nd: YVO4 laser (High Q Laser) was used to generate a 7 ps, 76 MHz pulse train of both 1,064 nm (1 W average power) and 532 nm (5 W average power) laser beams. The 1,064 nm output was used as the Stokes light. The 532 nm beam was 50/50 split to pump two optical parametric oscillators (Levante Emerald, A · P · E Angewandte Physik und Elektronik GmbH). The output wavelength of the optical parametric oscillators was at 909 nm to use as pump beam to induce the Stimulated Raman signal for the *1,600* cm$^{-1}$ lignin aromatic ring vibration. All pump and Stokes beams were directed into an Olympus laser scanning microscope scanning unit (BX62WI/FV300; Olympus) and focused by a high numerical aperture water-immersion objective (UPLSApo 60×1.20 NA W; Olympus). The light transmitted through the sample was collected by an oil-immersion condenser (1.45 NAO; Nikon). The stimulated Raman loss signals were detected by silicon PIN photodiodes (FDS1010; Thorlabs) and a lock-in amplifier (SR844; Stanford Research Systems)[54]. For each type of plant sample, at least three images were selected for intensity analysis. About 50 cells were selected and the average intensity for selected cells was calculated.

## In vitro transactivation assays

For the vector construction, *Pag-miR408* was driven by a constitutive 35 S promoter, no promoter (Nonpro) driving luciferase (LUC) was the negative control, and proActin2 driving LUC was the positive control. At the predicted binding site of target LACs and *miR408*, 5 nucleotides were mutated to remove the *miR408* recognition site, while at the same time guaranteeing that the amino acid sequences were unchanged. The mutated CDSs of *LAC19*, *LAC25* and *LAC32* were named *ΔLAC19*, *ΔLAC25*, and *ΔLAC32*. The complete CDSs of *LAC19*, *LAC25* and *LAC32*, and of *ΔLAC19*, *ΔLAC25*, and *ΔLAC32* were fused into pGreenII 0800 + pActin2 vector. *A. tumefaciens* GV3101 strains harboring the

different constructs were incubated in LB liquid medium containing 50 mg/L rifampicin and 50 mg/L kanamycin for 24 h to reach OD$_{600}$ = 1.0. The bacteria were spun down at 2,600 × $g$ for 5 min and the pellets were resuspended in infiltration buffer (10 mM MgCl$_2$, 10 mM MES pH 5.6, and 100 μM acetosyringone) with a final concentration of OD$_{600}$ = 0.5. The strain suspensions were kept at room temperature for 2 h before infiltration. For co-infiltration, equal volumes of two different strains carrying *miR408_OX* and *LAC-LUC* constructs were mixed prior to infiltration. The bacteria were infiltrated into *N. benthamiana* leaves using a 1 mL disposable syringe. After 72 h, the infiltrated leaves were sprayed with luciferin, and the fluorescence was detected by CCD camera (Vilber NEWTON7.0).

## Binding of CBM-GFP and cellulase to stem cross-sections

Basal stems of two-month-old tissue cultured poplars and one-year-old natural dried poplars of *miR408_OX* poplars were examined. A single-blade razor was used to prepare hand-cut transverse sections. All stem sections were examined under a bright field light microscope to select samples with relatively uniform cutting and a thickness of approximately 25-30 μm. The *Ct*CBM3-GFP contains an N-terminal 6xHis tag, a CBM, a short linker sequence (VPVEK) and a GFP. *Tr*CBM1 is a family 1 CBM, also containing a sequence encoding an N-terminal 6xHis tag. The C-terminal GFP tagged *Ct*CBM3 sequence was synthesized (Rui-Biotech) with genetic code optimization for expression in *Escherichia coli*. The synthesized DNA fragment was amplified by PCR using the forward primer *Ct*CBM3-F: 5'-GTTTAACTTTAAGAAGGAGATATACA-TATGACCATGATTACGCCAAGCTTGCATGCC-3', and the reverse primer *Ct*CBM3-R: 5'-GAAAAGTTCTTCTCCTTTACTCATTTTTTCTACC GGTACCAGGCACTGGGAGTAGTACG-3', *Tr*CBM1-F: 5'- GTTTAACTTTA AGAAGGAGATATACATATGACCATGATTACGCCAAGCTTGCATGCCC-3', *Tr*CBM1-R: 5'- CTCAGTGGTGGTGGTGGTGGTGCTCGAGTTATTTG-TAGAGCTCATCCATGCCATGTG-3' and inserted into pET21a vector using the seamless cloning method. The generated *Ct*CBM3-GFP and *Tr*CBM1-GFP expression plasmids were expressed in the *E. coli* strain BL21 (Novagen). Fusion proteins were purified through Ni-NTA column affinity chromatography (GE Healthcare). The *Ct*CBM3-GFP and *Tr*CBM1-GFP concentration was measured by NanoDrop™ 3300 (Thermo Scientific, Wilmington, DE, USA). The working concentration of purified *Ct*CBM3-GFP and *Tr*CBM1-GFP was 0.5 mg/mL[54], and the GFP-tagged proteins were stored in 20 mM Tris buffer (pH 8.0) at 4°C.

A commercial cellulase mixture was purchased from Yakult (Cellulase RS, Yakult, Japan). The cellulase was labeled according to the instructions supplied with the Alexa Fluor® 488 protein labeling kit (A10235, Invitrogen, Carlsbad, CA USA). For cell wall binding, 50 μL GFP-tagged *Ct*CBM3-GFP and *Tr*CBM1-GFP labeled cellulase were added to a piece of stem transverse section, incubated at room temperature for 30 min, and rinsed three times[54]. The labeled cell walls were imaged immediately by fluorescence microscopy (Leica DM2500, Germany). GFP and lignin were excited by a 488 nm laser and UV light, respectively. The fluorescence images were analyzed using Leica LAS X software.

## Determination of lignin molecular weight

One-year-old *miR408_OX* grown in the greenhouse natural debarked dried poplars stems were used as the plant materials. To isolate double enzymatic lignin (DELs), ball-milled poplar wood powder (5 g) was suspended in 100 mL acetate buffer (pH 4.8) with loading of 2.5 mL of cellulase (100 FPU/mL). The reaction mixture was incubated at 50 °C on a rotary shaker (150 rpm) for 48 h. After that, the mixture was centrifuged, and the residual solid was washed extensively with acetate buffer (pH 4.8) and hot water and then freeze-dried. The dried residual solid was further subjected to the ball-milling process for 2 h and enzymatic hydrolysis as in the abovementioned procedures. After washing and freeze-drying, the DEL fractions were obtained[55].

To improve the solubility of the lignin macromolecules in tetrahydrofuran for gel permeation chromatography (GPC) analysis, the lignin was acetylated[55]. 20 mg of lignin fractions were completely dissolved in DMSO/N-methylimidazole (2:1, v/v, 0.9 mL). After continually shaking for 24 h at 25 °C in the dark, acetic anhydride (0.6 mL) was added and the mixture incubated for 2 h. Finally, the solution was dropped slowly into ice water (pH 2.0, 100 mL) to precipitate the acetylated lignin samples, and the material centrifuged and freeze-dried.

The weight-average (Mw) and number-average (Mn) molecular weights of the acetylated lignin samples were determined by GPC. The GPC system (Agilent 1200, USA) was equipped with a UV detector (240 nm) and a PL-gel 10 μm Mixed-B 7.5 mm ID column, which was calibrated with monodisperse PL polystyrene standards (1390, 4830, 9970, 29150, 69650 g/mol). Four milligrams of acetylated lignin sample were dissolved in 2 mL of tetrahydrofuran (THF), and 10 μL of this solution was injected into a vial for testing. The column was operated at ambient temperature and eluted with THF at a flow rate of 1 mL min$^{-1}$.

## NMR sample preparation and NMR experiments

2D-HSQC NMR spectroscopy was performed on isolated (double enzymatic) lignin from stems of mature one-year-old *miR408_OX* grown in the greenhouse[56]. Fresh samples of 6-month-old stems of WT and *miR408_OX* plants were dried at 60 °C for 24 h and then stored in plastic bags before use. 5 g of each wood sample was smashed to obtain 20-40 mesh wood meal samples and was then milled in a planetary ball mill (Fritsch GmbH, Idar-Oberstein, Germany) at a fixed frequency of 500 rpm for 5 h. To prepare high-yield of double enzymatic lignin (DEL) from WT and *miR408_OX* poplar woods for NMR analysis, the ball-milled samples were subjected to double ball-milling and enzymatic hydrolysis according to the method as described in the first paragraph of "Determination of lignin molecular weight".

About 20 mg of DEL was dissolved in 0.5 mL of DMSO-$d_6$ (99.8% D). For quantitative 2D-HSQC spectra, the Bruker standard pulse program hsqcetgpsi2 was used. The spectral widths were 5000 Hz and 20000 Hz for the $^1$H- and $^{13}$C-dimensions, respectively. The number of collected complex points was 1024 for $^1$H-dimension with a recycle delay of 1.5 s. The number of transients was 64-and 256-time increments were always recorded in the $^{13}$C-dimension. The $^1J_{CH}$ used was 145 Hz. Prior to Fourier transformation, the data matrixes were zero filled up to 1024 points in the $^{13}$C-dimension. Data processing was performed using standard Bruker Topspin-NMR software.

The aromatic region of the S lignin and G lignin subunits was detected by the signals around 103.8/6.67 (S2/6), 106.4/7.38 (S'2/6), 110.5/6.94 (G2), 114.8/6.88 (G5) and 118.6/6.81 (G6) ppm, with the presence of their respective diagnostic correlation signals. A quantitative analysis of the intensities of the HSQC cross-signal was performed according to the following formula:

$$I(C_9) \text{ units} = 0.5I(S_{2,6}) + I(G_2) \tag{1}$$

where I(S$_{2,6}$) is the integration of S$_{2,6}$, including S and S', I(G$_2$) is the integral value of G$_2$, and I(C$_9$) represents the integral value of the aromatic ring. According to the internal standard (I(C$_9$)), the amount of I(*X*)% could be obtained by the following formula:

$$I(X)\% = I(X)/I(C_9) \times 100\% \tag{2}$$

where I(*X*) is the integral value of α-position of A (β-*O*−4), B (β-β), and C (β−5), respectively. The integration was performed in the same contour level. In the aromatic region, C$_{2,6}$-H$_{2,6}$ correlations from S units and C$_2$-H$_2$ correlation from G units were used to estimate the S/G ratio of lignin by the following formula[56]:

$$S/G = 0.5I(S_{2/6})/I(G_2) \tag{3}$$

## Reporting summary

Further information on research design is available in the Nature Portfolio Reporting Summary linked to this article.

## Data availability

The RNA-Seq data have been deposited in the Sequence Read Archive (SRA) with the accession number PRJNA967342. The authors declare all data generated or analyzed during this study are included in this published article and its Supplementary Information files. Source data are provided with this paper.

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

## Acknowledgements

This work was supported by the Program of Introducing Talents of Discipline to Universities (111 project, B13007 to J. L.), the National Natural Science Foundation of China and DFG (31761133009 to J. L.), China Postdoctoral Science Foundation funded project (2021M700450 and 2022T150054 to Y. G.), and the National Natural Science Foundation of China (31900173 to H.-L.W., 32000558 to X. Z.). We thank Dr. Yuanzhen Suo (Biomedical Pioneering Innovation Center, Peking University) for guidance regarding SRS microscopy, Dr. Ye Liang (State Key Laboratory of Membrane Biology, Peking University) for assistance with optical microscopy, Dr. Zhifang Wang

(Renishaw Public Limited Company) and Dr. Chunyang Wang (HORIBA Scientific) for Confocal-Raman analysis, and Dr. Pawan Kumar Jewaria (Beijing Forestry University) for insightful comments on the manuscript.

## Author contributions

J. L. and R. A. D. conceived the project; Y. G., S. W, J. L. and R. A. D. designed the experiments; Y. G., S. W., K. Y., H.-L.W., H. X., C. S., J. W., Yuanyuan Z., C. F., Yu L., S. W., X. W., X. D., Y. C., F. S., Y. Z. and Q. Z. carried out experiments; Y. G., S. W., K. Y., H.-L.W., H. X., C. S., J. W., X. Z., T. C., Yuanyuan Z., L. L., G. W., P. G., L. S., G. S., Y. L. and R.A.D. analyzed the data. Y. G., S. W., H.-L.W., R. A. D. and J. L. wrote the manuscript.

## Competing interests

The authors declare no competing interests.

## Additional information

[1]State Key Laboratory of Tree Genetics and Breeding, College of Biological Sciences and Technology, Beijing Forestry University, Beijing 100083, China. [2]Institute of Botany, Chinese Academy of Sciences, Beijing 100093, China. [3]College of Biological Sciences, China Agricultural University, Beijing 100193, China. [4]College of Agriculture, Henan University of Science and Technology, Luoyang 471003, China. [5]Beijing Key Laboratory of Lignocellulosic Chemistry, Beijing Forestry University, Beijing 100083, China. [6]Qingdao Institute of Bioenergy and Bioprocess Technology, Chinese Academy of Sciences, Qingdao 266101, China. [7]State Key Laboratory of Tree Genetics and Breeding, Chinese Academy of Forestry, Beijing 100091, China. [8]Shenzhen Key Laboratory of Synthetic Genomics, Shenzhen Institutes of Advanced Technology, Chinese Academy of Sciences, Shenzhen 518055, China. [9]School of Life Sciences and School of Advanced Agricultural Sciences, Peking University, Beijing 100871, China. [10]National Centre for Plant Gene Research (Beijing), Institute of Genetics and Developmental Biology, Chinese Academy of Sciences, Beijing 100101, China. [11]Department of Ecophysiology, Institute of Cellular and Molecular Botany, University of Bonn, Kirschallee 1, 53115 Bonn, Germany. [12]BioDiscovery Institute and Department of Biological Sciences, University of North Texas, Denton, TX 76203, USA. [13]These authors contributed equally: Yayu Guo, Shufang Wang. [14]These authors jointly supervised this work: Richard A. Dixon, Jinxing Lin. ✉e-mail: Richard.Dixon@unt.edu; linjx@ibcas.ac.cn

