## [Peer Review File · Nature Communications]

Manipulating non-coding RNA enhances both biomass yield and saccharification efficiency in poplarReviewers' Comments:

Reviewer #1:

Remarks to the Author:

This is a resubmission of a previously reviewed manuscript (NCOMMS-20-43668). In this new submission, the authors have made several important revisions with substantial new data to address the comments raised during the previous review. I particularly appreciate the efforts generating and characterizing overexpression and knockout out poplar lines for LAC genes to provide genetic evidence supporting that these LAC genes are targets of miR408 and work in the same pathway to regulate plant growth and saccharification efficiency. Overall, the quality of the present manuscript has been much improved over the previous version. There are several concerns that the authors need to address:

1) MiR408 targets: In the previous version, LAC19/20 were shown to be the targets of MiR408 whereas in the present version, LAC19/25/32 were shown to be the targets. I am just curious about the discrepancy. In vitro transactivation assays were used in both versions. Some explanations about the potential cause of discrepancy would be helpful. I did comment in the previous version that the examination was not comprehensive since only LAC19/20 were selected for the in vitro transactivation assays and other LACs (LAC16/25/32) were not. I am glad that the authors addressed my previous comments which has led discovery of new targets (LAC25/32), but now I wonder how LAC20 is no longer a target.

2) Determination of miR408 targets: the authors used RNA-Seq data from miR408-OX plants to determine DEGs and subsequently used ontology clustering to narrow down down-regulated genes/pathways. The authors then used psRNAtarget prediction to identify LAC19/25/32 as the highest potential targets of miR408. Since LAC47/55 were also predicted to be targets of MiR408 (Table S2) and given that they are in the same LAC family, I wonder why LAC47/55 (again, for comprehensiveness) were not included in the RT-PCR analysis, in vitro transactivation assays and 5' RACE. It is nice to see all positive data on LAC19/25/32, but it is equally important to see negative data on LAC47/55 to demonstrate specificity.

3) FigS14: What do DeltaLAC19/25/32 stand for? I understand there is some information about it in the Methods section (i.e., "five bases were mutated"; which five bases?), but this needs to be clearly defined in the figure legend. It was indicated that "The low panels show bright field photographs" but I don't see such photographs in panel b.

4) S/G ratio: miR408-OX plants have increased saccharification efficiency with decreased lignin content and also decreased S/G ratio. High S/G ratios are typically associated with increased saccharification efficiency. Can the authors elaborate a bit more or provide possible explanation about the opposite S/G ratio association observed in the present study?

5) Consistence in nomenclatures: MIR408 or miR408? It shows up in many places.

6) Fig 1: In the text, it states that two CRISPR lines of miR408 (#8 and #20) were selected for further study but in Fig 1, there are 3 lines (#8/18/20). It is not a major issue, but consistency would be nice.
Jin-Gui Chen

Reviewer #2:

Remarks to the Author:

Woody biomass is one of the most important sources of renewable energy around the world. In the study, the authors showed that overexpression of an miRNA, Pag-miR408 in hybrid poplar can significantly improve plant growth and also saccharification efficiency without acid pretreatment. They further validated that three laccases genes, Pag-LAC19, Pag-LAC25 and Pag-LAC32, are the directly targets of Pag-miR408 in hybrid poplar. LACCASES loss of function mutants also showed significantly enhanced growth and increased saccharification efficiency, and the cell wall deconstruction phenotypes observed are likely the result of laccases down-regulation. The findings revealed the mechanism of Pag-miR408 in lignification and secondary growth, and represent an effective approach towards enhancing biomass and producing lignocellulosic bioenergy for sustainable development.

Overall, the study was well designed and the experiments were well performed, especially those phenotypic analysis and localization investigation of LACs and Pag-miR408 using fluorescence in situ hybridization; the writing is also easy to follow and understand. I only have a few comments as follows, these issues need to be addressed before the manuscript can be accepted:

1. Title "Manipulating non-coding RNA....." better changes to "Manipulating an microRNA miR408..." or "Manipulating Pag-miR408....".

2. In the Introduction section, the authors only generally described lignocellulosic biomass and the approaches to overcoming the growth defects in lignin-modified plants, the authors should introduce why choosing and focusing miR408.

3. Figure 1d: MIR408-OX plants clearly showed enlarged vascular cambium zones, suggesting that overexpression of miR408 not only enhances biomass yield and saccharification efficiency, but also promotes cambium division, can authors discuss what are the underlying mechanism for miR408 to regulate cambium?

4. Figure 3a. The basal stem xylem width, also the stem width of MIR408-OX plants indeed showed significantly increased compared to WT, while no significant change occurred in knockout poplars (Figure 3a, Supplemental Figure 9a). Generally, knockout mutants should show an opposite phenotype with that of OX plants, can authors explain this?

5. Figure 5. Double lac25 lac32 and triple lac19 lac25 lac32 mutants showed lighter lignin staining, and loose cell arrangement with a degree of vessel collapse (figure 5b and c), and the cell wall morphology of the single gene mutants of lac19 was not as obvious as that of the triple and double mutants (Figure 5d), however, Figure 5d showed that the lac19 plants grow well as those of double and triple mutants, can authors explain these phenomenon?

In addition, LAC19-OX, LAC25-OX and LAC32-OX (Figure 5f-h) exhibited neatly arranged xylem cells, with similar morphology to WT. If these laccases indeed the targets of miR408, the LACs OX plants should show an opposite phenotype with those of miR408 OX plants, and the staining of xylem cells should be stronger than that of WT?

6. In the discussion section, could authors discuss more the function of laccases?

Reviewer #3:

Remarks to the Author:

This manuscript describes the phenotypes of transgenic poplar trees expressing a microRNA (miR408) that represses expression of genes encoding laccases. A triple mutant of three laccase-encoding genes phenocopies the miR408-expressing lines. Lignin content and composition are not substantially affected in these lines. However, the yield of glucose in saccharification assays is increased when the materials are treated with cellulase enzymes. There is an increased abundance of binding sites in transverse sections of these materials for probes that bind to cellulose microfibrils compared to wild type.

These results are significant in a biotechnological context of the deconstruction of lignocellulosic biomass as a source of sugars for conversion to fuels or other products. As the authors point out, reduction of lignin content is generally associated with reduced growth, an undesirable phenotype in a bioenergy crop. There are some examples in the literature, however, where changing lignin composition does not impact growth, for example, by expressing ferulate-5-hydroxylase in poplar trees to synthesize predominantly syringyl (S)-lignin. S-lignin is a linear molecule with a single type of

linkage that is more labile than the multiple kinds of linkages between aromatic subunits found in wild type lignin, comprising guaiacyl (G)- and S-lignin monomers. By contrast, the miR408-expressing lines and the triple laccase mutant described in this manuscript show slightly reduced S-lignin composition (Supplemental Table 1) compared to wild type, and about a 10% reduction in lignin content by AcBr assays.

Of more fundamental scientific interest, but very briefly noted in a single sentence in the Discussion, is the function of laccases in plant cell walls, and more specifically in lignin cross-linking. Laccases have previously been hypothesized to be involved in cross-linking monomeric subunits of lignin, and so, one interpretation of the authors' results is that reduced cross-linking accounts for the observed phenotypes.

This manuscript is rich in experimental results and there are few studies of such depth in transgenic tree species. However, the authors should address a number of issues in the writing of the manuscript before it is suitable for publication.

1. The authors should correct grammatical and typographical errors throughout.
2. Some terms are not precisely defined. For example, line 103, "strongly observed", line 107, 115 and Supplemental Figure 7, "loosely arranged xylem". In particular, the authors use the term "accessibility" to describe both macroscopic properties (glucose yields in saccharification assays) and microscopic properties (binding sites in transverse sections for cellulose-binding probes). I assume that the authors mean the accessibility of cellulose microfibrils to cellulase enzymes. It is reasonable to infer that both of these phenotypes are correlated with cellulase accessibility to its substrate but these are proxy measurements of accessibility rather than direct measurements.
3. Loss-of-function phenotypes from knockout of the miRNA-encoding gene are not described or discussed beyond a cursory mention in the text. However, Supplemental Figure 9b and c shows interesting phenotypes of reduced cell wall thickness and an increased intensity of phloroglucinol staining in successive internodes. Discussion of these phenotypes could enrich understanding of the function of miR408.
4. Throughout the results, the authors should clarify the nature of the samples measured. For example, in Figure 1i, Table 1, Supplemental Figure 4, Figure 4d, the legend states that $n = 3$, without specifying whether these represent 3 transgenic lines, 3 trees sampled from one transgenic line, or three replicate samples from a single tree.
5. Figure 4 also indicates that Lac 47 and Lac 55 are potential targets of miR408 but no data are presented – are there changes in the expression levels of these two laccases? Supplemental Figure 12c shows altered expression ratios of Laccases 1, 3, 4, 10, 11 and 17, but not 19, 25 or 32 in the miR408-expressing lines. Is this a typographical error? Or is the expression of these other laccase genes also impacted in the laccase triple mutant?
6. Line 267 refers to Figure 4e to g, but these panels are not part of Figure 4.
7. The authors should cite relevant literature and discuss their own findings in the context of literature with respect to the function of laccases in lignification. For example, a dirigent protein (Dirigent protein 23) is also a predicted target gene of miR408 (Supplemental Table 2), and this class of protein has also been implicated in lignin cross-linking. How is its expression affected in the triple laccase mutant and does this impact (or not) the interpretation of the triple mutant phenotype?
8. Supplemental Figure 8a – please clarify if "cell wall residues" refers to cell walls after cellulase treatment, or if this is the total sugar content of the starting cell wall materials.
9. Supplemental Figure 14b legend, where are the "lower panels" showing bright field photographs?

Response to Reviewer 1

Comments of Reviewer 1:

This is a resubmission of a previously reviewed manuscript (NCOMMS-20-43668). In this new submission, the authors have made several important revisions with substantial new data to address the comments raised during the previous review. I particularly appreciate the efforts generating and characterizing overexpression and knockout out poplar lines for LAC genes to provide genetic evidence supporting that these LAC genes are targets of miR408 and work in the same pathway to regulate plant growth and saccharification efficiency. Overall, the quality of the present manuscript has been much improved over the previous version. There are several concerns that the authors need to address.

Point 1: MiR408 targets: In the previous version, *LAC19/20* were shown to be the targets of MiR408 whereas in the present version, *LAC19/25/32* were shown to be the targets. I am just curious about the discrepancy. In vitro transactivation assays were used in both versions. Some explanations about the potential cause of discrepancy would be helpful. I did comment in the previous version that the examination was not comprehensive since only *LAC19/20* were selected for the in vitro transactivation assays and other *LACs* (*LAC16/25/32*) were not. I am glad that the authors addressed my previous comments which has led discovery of new targets (*LAC25/32*), but now I wonder how *LAC20* is no longer a target.

Response 1: Sorry for the unclear explanation here. After receiving your comments on the previous version, we performed detailed new experiments to re-address the targets. In the last two years since the first submission, we have established the 5' RACE technique to confirm the true targets of *miR408* in 84K poplar.

- In the first version, we obtained only *miR408_OX* materials without *miR408_cr*. We found five differentially expressed laccases, four were downregulated (*Potri.008G073800, LAC19; Potri.010G183500, LAC25; Potri.013G152700, LAC32; Potri.009G034500, LAC20*) and one was upregulated (*Potri.007G023300, LAC16*) from the first-time RNA-Seq data between *miR408_OX* and WT.
- As for *LAC16*, we found from the first-time RNA-Seq data that *LAC16* was upregulated. Since microRNAs play a negative regulatory role after transcription, *LAC16* could not be the target gene of *miR408*. On the other hand, using psRNA target prediction, we also

found that *LAC16* was not present in the potential targets list of *miR408* (Supplemental table 2). Therefore, we can safely conclude *LAC16* is not the target of *miR408*.

- In the second version of the paper, based on the fact that we have obtained the *miR408-cr* lines, we re-performed the RNA-seq using WT, *miR408_OX* and also *miR408_cr*. The transcript data showed the same trend in duplicate analyses; the expression of *LAC19*, *25* and *32* was all down-regulated in the over-expression lines. Furthermore, from psRNAtarget prediction, *LAC19*, *LAC25*, and *LAC32* were predicted as the highest potential targets of *miR408* (Supplemental table 2). This is the main reason why we selected *LAC19*, *25* and *32* to verify our analyses. We did not find that *LAC20* was present in the potential targets list of *miR408* and there is no potential cleavage site of *LAC20* that can be targeted by *miR408*. Moreover, we did not find differences in the expression level of *LAC20* in the second RNA-seq data. Based on these results, we are confident that *LAC20* is not the target of *miR408* and therefore deleted it in the revised version. Thanks for your comments which led to discovery of new targets (*LAC25/32*) in from our additional experiments.

Point 2: Determination of *miR408* targets: the authors used RNA-Seq data from *miR408_OX* plants to determine DEGs and subsequently used ontology clustering to narrow down down-regulated genes/pathways. The authors then used psRNAtarget prediction to identify *LAC19/25/32* as the highest potential targets of *miR408*. Since *LAC47/55* were also predicted to be targets of *miR408* (Table S2) and given that they are in the same *LAC* family, I wonder why *LAC47/55* (again, for comprehensiveness) were not included in the RT-PCR analysis, in vitro transactivation assays and 5' RACE. It is nice to see all positive data on *LAC19/25/32*, but it is equally important to see negative data on *LAC47/55* to demonstrate specificity.

Response 2: After reading your comments, we realized this problem in our previous version. To solve this, we carried out qRT-PCR and 5' RACE for the identification of target genes. We first analyzed the down-regulated laccases in transcriptome data, and also checked whether the down-regulated laccases exist in the list of predicted target genes. If down-regulated laccases do not exist in the predicted target gene list, it means that these laccases have no target cleavage site for *miR408*. We believe these down-regulated laccases are not the

targets of *miR408*. Therefore, we selected *LAC19*, *25* and *32* to do 5' RACE to find the cleavage site of *miR408*.

The results showed that *LAC47* and *55* were not in the list of down-regulated genes in both transcriptomic analyses (Figure S12d). Furthermore, in our qRT-PCR analysis, we found that there were no significant differences in the expression level of *LAC47* and *55* between WT and *miR408_OX* (Figure S12e). Although *LAC47* and *55* were found in the predicted target gene list, the predicted score is rather low, indicating that the interaction between *miR408* and *LAC47* and *55* is likely very weak. Nevertheless, we carried out 5' RACE to find whether *miR408* can cleave *LAC47* and *55*. Although bands of the predicted size could be amplified (Figure SS1), we found from sequencing data that there was no cleavage site in *LAC47* and *55* by *miR408*. Taken together, we are confident that *LAC47* and *55* are not real targets of *miR408*. Accordingly, we added these results in Figure S12d-e and some sentences were added in the Results section.

Supplemental Figure 12. Transcriptomic and qRT-PCR analyses of 3-month-old WT and *miR408_OX* and knockout plants.

Figure SS1. Agarose gel electrophoresis showing the 5' RACE products and size markers (M). (This data is not displayed in the manuscript or supplemental materials.)

Point 3: FigS14: What do DeltaLAC19/25/32 stand for? I understand there is some information about it in the Methods section (i.e., “five bases were mutated”; which five bases?), but this needs to be clearly defined in the figure legend. It was indicated that “The low panels show bright field photographs” but I don’t see such photographs in panel b.

Response 3: We are sorry for the unclear descriptions in the figure legend. Now we have added the experimental details and descriptions to the method of *in vitro* transactivation assays, “At the predicted binding site of target *LACs* and *miR408*, 5 nucleotides were mutated to remove the *miR408* recognition site, while at the same time guaranteeing that the amino acid sequences were unchanged. The mutated CDSs of *LAC19*, *LAC25* and *LAC32* were named $\Delta LAC19$, $\Delta LAC25$, and $\Delta LAC32$.”

- We also clearly defined in the figure legend that “5 nucleotides were mutated at the predicted binding site of target *LACs* and *miR408*, in order to disrupt the *miR408* recognition site, while at the same time guaranteeing that the amino acid sequences were unchanged. The mutated CDSs of *LAC19*, *LAC25* and *LAC32* were named $\Delta LAC19$, $\Delta LAC25$, and $\Delta LAC32$. The original predicted binding sites of *LAC19*, *LAC25* and *LAC32* were TCCAGTGAAGAGGCTGTGCAA, TCCAGTGAAGAGGCTGTGCAA and ACCAGTGAAGAGGCTGTGCAG, respectively, and the mutated binding sites of $\Delta LAC19$, $\Delta LAC25$, and $\Delta LAC32$ were TCCGGTAAAAAGACTGTGTAA, TCCGGTAAAAAGACTCTGTAA and ACCGGTAAAAAGACTGTGTAG, respectively.” We annotated this on Figure S14a.

- We apologize for missing the bright field photographs. They have now been added as Figure S14b.

Supplemental Figure 14. Functional identification of targets of miR408 in planta.

Point 4: S/G ratio: *miR408_OX* plants have increased saccharification efficiency with decreased lignin content and also decreased S/G ratio. High S/G ratios are typically associated with increased saccharification efficiency. Can the authors elaborate a bit more or provide possible explanation about the opposite S/G ratio association observed in the present study?

Response 4: Thanks for your suggestion. It has been previously reported that reducing lignin content/altering the lignin composition and increasing accessibility of cellulose microfibrils to cellulase enzymes can be used to improve biomass digestibility of cell walls (Cassie *et al.*, 2015). In our study, the bulk lignin in *miR408_OX* poplar exhibited modest reductions in total content, S/G ratio, and degree of polymerization. A larger overall decrease in lignin content than the 4-10% recorded in the present work is usually needed to enhance cell wall saccharification (Studer *et al.*, 2011), but decreases in secondary xylem cells likely were higher. The decreased lignin S/G ratio results in more condensed lignin, although its impact

on recalcitrance to acid pre-treatment depends on the lignin content and additional factors in poplar (Davison et al., 2006). Higher S/G ratios have been reported to have a positive effect on biomass saccharification after hot-water pre-treatment in poplar (Franke et al., 2000; Studer et al., 2011; Nawawi et al., 2017), suggesting that the decreased S/G ratio in *miR408_OX* plants is unlikely to contribute to the greatly enhanced sugar release. In our study, we found increased accessibility of vascular tissues to *C. thermocellum* CBMs and fungal cellulase in *miR408_OX* plants, demonstrating that cell walls may be more “open” to deconstruction in *miR408_OX* plants.

Cassie, W, Vimal, B, Carloalberto, P, Krishan, R, Seth, DB, Venugopal, M (2015). Engineering plant biomass lignin content and composition for biofuels and bioproducts. *Energies*, 8(8), 7654-7676.

Studer MH, DeMartini JD, Davis MF, Sykes RW, Davison B, Keller M, Tuskan GA, Wyman CE (2011). Lignin content in natural *Populus* variants affects sugar release. *Proc. Natl. Acad. Sci. U S A*, 108(15):6300-5.

Franke, R, Mcmichael, CM, Meyer, K, Shirley, AM, Cusumano, JC, Chapple, C (2000). Modified lignin in tobacco and poplar plants over-expressing the Arabidopsis gene encoding ferulate 5-hydroxylase. *Plant J.* 22(3):223-34.

Nawawi, D, S, Syafii, W., Tomoda, I., Uchida, Y., Akiyama, T., Yokoyama, T., and Matsumoto, Y. (2017). Characteristics and reactivity of lignin in *Acacia* and *Eucalyptus* woods. *J. Wood Chem. Technol.* 37: 273-282.

Point 5: Consistence in nomenclatures: MIR408 or miR408? It shows up in many places.

Response 5: Sorry for some inconsistencies in nomenclature. After referring to the paper in Plant Cell (Jiang et al, 2021), we changed all “MIR408” in the paper including in the manuscript and figures to “*miR408*” in order to ensure consistency, in which “MIR408” and “*miR408*” refer to the precursor sequence of miRNA, while “miR408” refers to the mature sequence of 21bp that can bind the cleavage site of the target genes.

Jiang A, Guo Z, Pan J, Yang Y, Zhuang Y, Zuo D, Hao C, Gao Z, Xin P, Chu J, Zhong S, Li L. The PIF1-miR408-PLANTACYANIN repression cascade regulates light-dependent seed germination (2021). *Plant Cell* 33(5):1506-1529.

Point 6: Fig 1: In the text, it states that two CRISPR lines of *miR408* (#8 and #20) were selected for further study but in Fig 1, there are 3 lines (#8/18/20). It is not a major issue, but consistency would be nice.

Response 6: Yes, we agree. In fact, we selected CRISPR lines of *miR408* (#8 and #20) for further analysis, after statistical analysis of plant height and stem diameter in three lines. Now we reworded this sentence as “After statistical analysis of plant height and stem diameter in three independent *miR408_OX* lines (1, 5, and 6) (Supplemental Figure 1c), two independent homozygous lines which had 218bp genomic deletions (*miR408_cr* #8 and #20) were selected for further study”.

Response to Reviewer 2

Comments of Reviewer 2:

Woody biomass is one of the most important sources of renewable energy around the world. In the study, the authors showed that overexpression of an miRNA, *Pag-miR408* in hybrid poplar can significantly improve plant growth and also saccharification efficiency without acid pretreatment. They further validated that three laccases genes, *Pag-LAC19*, *Pag-LAC25* and *Pag-LAC32*, are the directly targets of *Pag-miR408* in hybrid poplar. *LACCASES* loss of function mutants also showed significantly enhanced growth and increased saccharification efficiency, and the cell wall deconstruction phenotypes observed are likely the result of laccases down-regulation. The findings revealed the mechanism of *Pag-miR408* in lignification and secondary growth, and represent an effective approach towards enhancing biomass and producing lignocellulosic bioenergy for sustainable development.

Overall, the study was well designed and the experiments were well performed, especially those phenotypic analysis and localization investigation of *LACs* and *Pag-miR408* using fluorescence in situ hybridization; the writing is also easy to follow and understand. I only have a few comments as follows, these issues need to be addressed before the manuscript can be accepted:

Point 1. Title "Manipulating non-coding RNA....." better changes to "Manipulating an microRNA *miR408*..." or "Manipulating *Pag-miR408*....".

Response 1: Thanks for the helpful suggestions. We have replaced the title with "Manipulating microRNA *miR408* enhances both biomass yield and saccharification efficiency in poplar" as suggested.

Point 2. In the Introduction section, the authors only generally described lignocellulosic biomass and the approaches to overcoming the growth defects in lignin-modified plants, the authors should introduce why choosing and focusing *miR408*.

Response 2: As suggested, we have added several sentences to explain why we selected and focused on *miR408* in the "Introduction" section.

"MicroRNAs (miRNA), endogenous small noncoding RNAs of 21–24 nucleotides in length, are key eukaryotic gene regulators that play critical roles in plant development and stress tolerance (Brodersen and Voinnet, 2009), and some have targets that may be involved in

secondary wall formation and lignification (Hou *et al.*, 2020; Sun *et al.*, 2018). *miR408* is a highly conserved miRNA of 21 nucleotides, first identified in *Arabidopsis thaliana* (Axtell and Bowman, 2008). Its over-expression increases biomass and seed yield in *Arabidopsis* and rice, potentially through effects on copper-containing proteins plantacyanin and laccase (Pan *et al.*, 2018). These growth effects, coupled with the suggestion of *LACCASE* as a target, suggests the possibility that manipulating *miR408* might overcome the growth defects caused by the down-regulation of lignin synthesis through direct targeting of structural genes.”

Brodersen, P., and Voinnet, O. (2009). Revisiting the principles of microRNA target recognition and mode of action. *Nat. Rev. Mol. Cell Biol.* 10: 141-148.

Hou, J., Xu, H., Fan, D., Ran, L., Li, J., Wu, S., Luo, K., and He, X.Q. (2020). MiR319a-targeted PtoTCP20 regulates secondary growth via interactions with PtoWOX4 and PtoWND6 in *Populus tomentosa*. *New Phytol.* 228: 1354-1368.

Sun, Q., Liu, X., Yang, J., Liu, W., Du, Q., Wang, H., Fu, C., and Li, W.X. (2018). MicroRNA528 affects lodging resistance of maize by regulating lignin biosynthesis under nitrogen-luxury conditions. *Mol. Plant.* 11: 806-814.

Axtell, M.J., and Bowman, J.L. (2008). Evolution of plant microRNAs and their targets. *Trends Plant Sci.* 13: 343-349

Pan, J., Huang, D., Guo, Z., Kuang, Z., Zhang, H., Xie, X., M, Z., Gao, S., Lerda, M.T., Chu, C., and Li, L. (2018). Overexpression of microRNA408 enhances photosynthesis, growth, and seed yield in diverse plants. *J. Integr. Plant Biol.* 60: 323-340.

Point 3: Figure 1d: *miR408_OX* plants clearly showed enlarged vascular cambium zones, suggesting that overexpression of *miR408* not only enhances biomass yield and saccharification efficiency, but also promotes cambium division, can authors discuss what are the underlined mechanism for *miR408* to regulate cambium?

Response 3: Thanks for your suggestion. We have added the discussion about *miR408* in regulating cambium division in the “Discussion” section.

“Overexpressing *miR408* in poplar contributed the enhanced plant growth associated with a significant increase in net photosynthetic rate, somewhat similar to the phenotype when overexpressing *miR408* in *Arabidopsis* (Pan *et al.*, 2018). In addition, we found *miR408_OX* poplar showed wider cambium zone and increased xylem area (with enlarged xylem cells). We also showed that *miR408* was mainly expressed in the vascular cambium and developing xylem. These results suggest a specific function for *miR408* in wood

formation, which is characterized by sequential differentiation of vascular cambial cells into xylem cells, cell expansion, massive deposition of secondary cell walls, programmed cell death, and ultimately, the formation of heartwood (Zhang *et al.*, 2014; Wang *et al.*, 2020). Although these anatomical features were clearly linked to *miR408* overexpression or knock-out, more work is necessary to understand the coordination of the developmental and biochemical changes.”

Pan, J., Huang, D., Guo, Z., Kuang, Z., Zhang, H., Xie, X., M, Z., Gao, S., Lerda, M.T., Chu, C., and Li, L. (2018). Overexpression of microRNA408 enhances photosynthesis, growth, and seed yield in diverse plants. *J. Integr. Plant Biol.* 60: 323-340.

Zhang, J., Nieminen, K., Serra, J.A. & Helariutta, Y (2014). The formation of wood and its control. *Curr. Opin. Plant Biol.* 17, 56-63.

Wang, L. et al (2020). Multifunctional analyses of vascular cambial cells reveal longevity mechanisms in old Ginkgo biloba trees. *Proc Natl Acad Sci U S A* 117, 2201-2210.

Point 4. Figure 3a. The basal stem xylem width, also the stem width of *miR408_OX* plants indeed showed significantly increased compared to WT, while no significant change occurred in knockout poplars (Figure 3a, Supplemental Figure 9a). Generally, knockout mutants should show an opposite phenotype with that of OX plants, can authors **explain** this?

Response 4: Yes, there is indeed no opposite phenotype in the plant height and stem diameter between *miR408_OX* and *miR408_cr*. However, statistical analysis showed reduced numbers of lignified xylem cell layers in *miR408_OX*, while increased numbers were seen in *miR408_cr* compared with WT (Fig S9c). These results suggested overexpression of *miR408* can delay lignification, while knock out of *miR408* can promote lignification. We added explanations in the results part “***miR408* targets *PagLAC19*, *PagLAC25* and *PagLAC32***” as follows: “Using psRNA target prediction, three *LACCASES*, *LAC19*, *LAC25*, *LAC32* were predicted as the highest potential targets of *miR408* (Figure 4c, Supplemental table 2). qRT-pCR results showed *LAC19*, *LAC25* and *LAC32* transcripts were significantly decreased in *miR408_OX* plants while not obviously changed in *miR408_cr* lines (Figure 4d), mainly because all the five predicted *LACs*, namely *LAC19*, *LAC25*, *LAC32*, *LAC47* and *LAC55* were also predicted to be targeted by other microRNAs such as miR475, miR396, miR1447, miR397, miR169, miR7826, miR7466 and miR7817 (Figure 4c).” This result indicates that *LAC19*, *LAC25* and *LAC32* can also be targeted by other microRNAs. In the *miR408* mutant, to compensate for the lack of function of *miR408*, the plants may be able to generate other microRNAs to target and down-regulate *LAC19*, *LAC25* and *LAC32*, so as to

ensure that copper ions can be preferentially distributed to key copper-containing proteins (Zhang et al., 2014). Therefore, it is reasonable to believe that *lac* knockout mutants may not always show an opposite phenotype to that of overexpression poplars in plant height and stem diameter.

Zhang H, Zhao X, Li J, Cai H, Deng XW, Li L. (2014) MicroRNA408 is critical for the *HY5-SPL7* gene network that mediates the coordinated response to light and copper. *Plant Cell*. 26(12):4933-53.

Point 5. Figure 5. Double *lac25 lac32* and triple *lac19 lac25 lac32* mutants showed lighter lignin staining, and loose cell arrangement with a degree of vessel collapse (figure 5b and c), and the cell wall morphology of the single gene mutants of *lac19* was not as obvious as that of the triple and double mutants (Figure 5d), however, Figure 5d showed that the *lac19* plants grow well as those of double and triple mutants, can authors explain these phenomenon?

In addition, *LAC19_OX*, *LAC25_OX* and *LAC32_OX* (Figure 5f-h) exhibited neatly arranged xylem cells, with similar morphology to WT. If these laccases indeed the targets of *miR408*, the LACs OX plants should show an opposite phenotype with those of *miR408* OX plants, and the staining of xylem cells should be stronger than that of WT?

Response 5: Thanks for your suggestion. From Figure 5b-d, we can indeed observe that *lac19* plants can grow slightly higher than WT, and there is no significant difference from the statistical analyses. Compared with WT, the phenotypes of double *lac25 lac32* and triple *lac19 lac25 lac32* are more significant. The possible reason is that there are 55 members in the laccase family, and the lack of a single *lac19* has no significant impact on the phenotype.

- The *LAC19_OX*, *LAC25_OX* and *LAC32_OX* plants show decreased plant height and stem diameter compared with WT (Figure 5e-h). These phenotypes with reduced growth are opposite to those with enhanced growth in *miR408_OX*.
- After reading your comment, we realized that additional analysis would be helpful. To obtain accurate lignin content, we carried out a new experiment to measure the lignin content of WT, *LAC19_OX*, *LAC25_OX* and *LAC32_OX* using the Klason method as described in Chen et al (2017). From Table S4, the lignin contents of the *LAC19_OX*, *LAC25_OX* and *LAC32_OX* plants have a -21.56%, -11% and -14.62% increase, respectively. We also repeated the phloroglucinol staining assay of these lines and the

new results were similar to the chemical data, showing stronger staining than that of WT. Based on the new experimental results, we used the new staining photos in Figure 5.

Chen, T.-Y. *et al* (2017) Structural variations of lignin macromolecule from different growth years of Triploid of *Populus tomentosa* Carr. *Int. J. Biol Macromol.* 101, 747-757.

Figure 5. *lac19 lac25 lac32* mutant poplar shows enhanced growth.

Supplemental Table 4. Cell wall lignin content of WT and *LACs_OX* poplars.

	Lignin		Total lignin
	AIL	ASL	
WT	23.13 ± 0.32	4.70 ± 0.20	27.83 ± 0.35
LAC19_OX	29.53 ± 0.35	3.77 ± 0.06	33.31 ± 0.29
LAC25_OX	27.00 ± 0.26	3.80 ± 0.05	30.80 ± 0.31
LAC32_OX	28.07 ± 0.55	3.83 ± 0.06	31.90 ± 0.61

Contents of acid insoluble lignin (AIL), acid soluble lignin (ASL), total lignin. Values are means ± SE (n = 3, n represents 3 trees sampled respectively from each transgenic line). Values are expressed as weight percent based on vacuum-dried extractive free wood weight (% w/w).

Point 6. In the discussion section, could authors discuss more the function of laccases?

Response 6: Thanks for the helpful suggestion. We have added discussion about laccase function in the “Discussion” section.

Overexpression of *miR408* results in a large increase in saccharification efficiency with no requirement for acid-pretreatment for both laboratory- and field-grown poplar plants. We found that overexpression of *Pag-miR408* targets *Pag-LAC19,25,32*, delays lignification, and modestly reduces lignin content, S/G ratio and degree of lignin polymerization. The *lac* triple mutants showed similar phenotypes in vascular cell morphology, cell wall accessibility, and saccharification to *miR408_OX* poplars. Laccases are considered to function in the polymerization of lignin monomers, potentially at the stage of polymer initiation, and subsequently in concert with peroxidases³⁹. Given that reduced degree of lignin polymerization is associated with improved lignin extractability and reduced biomass recalcitrance⁴⁰, we speculate that the changes in lignin distribution and composition observed in the present *miR408_OX* plants result largely from the post-transcriptional regulation of the three target *LACCASES*. Both decreased lignin polymer size and delayed lignification in *miR408_OX* indicate a mechanistic basis for the improved saccharification efficiency, which is enhanced by the altered ultrastructure of the vascular tissues, brought about by delayed secondary cell wall deposition during development. As shown by the increased accessibility of vascular tissues to *C. thermocellum* CBMs and fungal cellulase, we can conclude that the high saccharification efficiency of *miR408_OX* plants is largely linked to the more “open” cell walls for (downstream) deconstruction process.

- Zhao, Q., Nakashima, J., Chen, F., Yin, Y., Fu, C., Yun, J., Shao, H., Wang, X., Wang, Z.Y., and Dixon, R.A. (2013). Laccase is necessary and nonredundant with peroxidase for lignin polymerization during vascular development in *Arabidopsis*. *Plant Cell*. 25: 3976-3987.
- Ziebell, A., Gracom, K., Katahira, R., Chen, F., Pu, Y., Ragauskas, A., Dixon, R.A., and Davis, M. (2010). Increase in 4-coumaryl alcohol units during lignification in alfalfa (*Medicago sativa*) alters the extractability and molecular weight of lignin. *J. Biol. Chem.* 285: 38961-38968.

Response to Reviewer 3

Comments of Reviewer 3:

This manuscript describes the phenotypes of transgenic poplar trees expressing a microRNA (*miR408*) that represses expression of genes encoding laccases. A triple mutant of three laccase-encoding genes phenocopies the *miR408*-expressing lines. Lignin content and composition are not substantially affected in these lines. However, the yield of glucose in saccharification assays is increased when the materials are treated with cellulase enzymes. There is an increased abundance of binding sites in transverse sections of these materials for probes that bind to cellulose microfibrils compared to wild type.

These results are significant in a biotechnological context of the deconstruction of lignocellulosic biomass as a source of sugars for conversion to fuels or other products. As the authors point out, reduction of lignin content is generally associated with reduced growth, an undesirable phenotype in a bioenergy crop. There are some examples in the literature, however, where changing lignin composition does not impact growth, for example, by expressing ferulate-5-hydroxylase in poplar trees to synthesize predominantly syringyl (S)-lignin. S-lignin is a linear molecule with a single type of linkage that is more labile than the multiple kinds of linkages between aromatic subunits found in wild type lignin, comprising guaiacyl (G)- and S-lignin monomers. By contrast, the *miR408*-expressing lines and the triple laccase mutant described in this manuscript show slightly reduced S-lignin composition (Supplemental Table 1) compared to wild type, and about a 10% reduction in lignin content by AcBr assays.

Of more fundamental scientific interest, but very briefly noted in a single sentence in the Discussion, is the function of laccases in plant cell walls, and more specifically in lignin cross-linking. Laccases have previously been hypothesized to be involved in cross-linking monomeric subunits of lignin, and so, one interpretation of the authors' results is that reduced cross-linking accounts for the observed phenotypes.

This manuscript is rich in experimental results and there are few studies of such depth in transgenic tree species. However, the authors should address a number of issues in the writing of the manuscript before it is suitable for publication.

Point 1. The authors should correct grammatical and typographical errors throughout.

Response 1: Done.

Point 2. Some terms are not precisely defined. For example, line 103, “strongly observed”, line 107, 115 and Supplemental Figure 7, “loosely arranged xylem”. In particular, the authors use the term “accessibility” to describe both macroscopic properties (glucose yields in saccharification assays) and microscopic properties (binding sites in transverse sections for cellulose-binding probes). I assume that the authors mean the accessibility of cellulose microfibrils to cellulase enzymes. It is reasonable to infer that both of these phenotypes are correlated with cellulase accessibility to its substrate but these are proxy measurements of accessibility rather than direct measurements.

Response 2: We have made revisions to more precisely define the terms we are using. For the sentence on line 103-105, “GUS signal was detected in leaf veins (Supplemental Figure 3b-c) and was strongly observed in root vascular tissue (Supplemental Figure 3d), strong in the vascular cambium that will differentiate into xylem, but weak in mature xylem (Supplemental Figure 3e-g)”. The sentence has been changed into “GUS signal was detected in leaf veins (Supplemental Figure 3b-c) and in root vascular tissue (Supplemental Figure 3d). Promoter activity of *miR408* was mainly detected in the vascular cambium that will differentiate into xylem, but weak in mature xylem (Supplemental Figure 3e-g).”

- Agreed, for the sentence in line 107, 115 and Supplemental Figure 7, the term of “loosely arranged xylem” is inappropriate.
- For the sentence in line 107, “Semithin sections results showed cells in the secondary xylem of *miR408_OX* plants were arranged more loosely with significantly increased number of vascular cambium layers (Figure 1d, Supplemental Figure 4a), and there were more xylem cells with increased area (Supplemental Figure 4b-c) and vessels (Supplemental Figure 4d-e) than in WT.” The sentence has been changed into “Semithin sectioning showed that cells in the secondary xylem of *miR408_OX* plants were enlarged (Figure 1d). The cambium zone of *miR408_OX* lines was wider (Figure 1d) by about 42% (Supplemental Figure 4a). Compared with WT, *miR408_OX* showed more xylem cells, xylem area (Supplemental Figure 4b-c) and vessels (Supplemental Figure 4d-e).”

- For the sentence in line 115, “The loosely arranged xylem” has been changed into “To test whether the enlarged xylem cells might possess more loosely-organized cell walls, we utilized green fluorescent protein.....”.
- For the sentence in Supplemental Figure 7, “were also more loosely arranged” has been changed into “were also enlarged”.
- For the accessibility, we used the term following Ding et al (2012, Science). We agree that the term “accessibility” in the context of our work means “accessibility of cellulose microfibrils to cellulase enzymes”, thus we changed the title in line 113 into “Overexpression of *miR408* increases secondary cell wall accessibility to cellulase enzymes and enhances saccharification efficiency.”
- For the sentence in line 115, “The loosely arranged xylem might increase the accessibility of secondary cell walls to cellulase.” has been changed into “The enlarged xylem cells might increase secondary cell wall accessibility of cellulose microfibrils to cellulase enzymes”.
- For the sentence in line 124-125, “Similarly increased cell wall accessibility” has been changed into “Similarly increased cell wall accessibility of cellulose microfibrils to cellulase enzymes.....”.
- For the sentence in line 164-165, the title “Overexpression of *miR408* enhances biomass yield and saccharification efficiency in field-grown plants” has been changed into “Overexpression of *miR408* enhances biomass yield and accessibility of cellulose microfibrils to cellulase enzymes in secondary cell walls of field-grown plants”
- For the sentence on line 171, “increased cell wall accessibility to hydrolytic enzymes” has been changed into “increased cell wall accessibility of cellulose microfibrils to cellulase enzymes”.
- For the sentence in line 325, “cell wall accessibility” has been changed into “cell wall accessibility of cellulose microfibrils to cellulase enzymes”.
- For the sentence in line 337, “showed increased cellulase accessibility” has been changed into “showed increased cell wall accessibility of cellulose microfibrils to cellulase enzymes”.
- For the sentence in line 342, “was” has been changed into “were”.

- We also agree that the accessibility of cellulose microfibrils to cellulase enzymes is a proxy measurement of accessibility rather than a direct measurement. Accordingly, we carried out additional chemical analysis of saccharification efficiency of *lac* mutants (Table S5). The results of chemical saccharification assays accorded with the microscopic results.

Table S5. Analysis of saccharification efficiency of *laccase* mutant poplars grown in greenhouse.

Digestion time	Sample name	Peak area	Glucose concentration (mg/ml)	Total glucose released (mg)	Saccharification efficiency
24h	WT	15665.33	0.17	17.04	21.69%
24h	lac19 lac25 lac32 (#4)	33686.01	0.27	27.10	66.43%
24h	lac19 lac25 lac32 (#14)	28873.03	0.24	24.41	45.25%
24h	lac19 lac25 lac32 (#16)	35744.33	0.28	28.25	48.87%
24h	lac25 lac32 (#12)	37009.02	0.28	28.96	32.73%
24h	lac25 lac32 (#22)	24361.37	0.21	21.89	43.48%
24h	lac25 lac32 (#24)	23234.10	0.21	21.27	44.77%
24h	lac19 (#1)	21844.01	0.20	20.49	27.59%
24h	lac19 (#2)	21919.67	0.21	20.53	26.91%
48h	WT	24939.07	0.22	22.22	28.28%
48h	lac19 lac25 lac32 (#4)	43166.33	0.32	32.40	79.41%
48h	lac19 lac25 lac32 (#14)	34840.01	0.28	27.74	51.43%
48h	lac19 lac25 lac32 (#16)	42563.07	0.32	32.06	55.46%
48h	lac25 lac32 (#12)	44735.02	0.33	33.27	37.61%
48h	lac25 lac32 (#22)	35295.33	0.28	27.99	55.60%
48h	lac25 lac32 (#24)	29899.11	0.25	24.97	52.60%
48h	lac19 (#1)	28858.12	0.24	24.41	32.87%
48h	lac19 (#2)	31822.33	0.26	26.06	34.16%
72h	WT	26214.67	0.23	22.93	29.19%
72h	lac19 lac25 lac32 (#4)	43601.05	0.33	32.64	80.00%
72h	lac19 lac25 lac32 (#14)	36420.33	0.29	28.63	53.06%
72h	lac19 lac25 lac32 (#16)	53946.33	0.38	38.41	66.45%
72h	lac25 lac32 (#12)	48086.67	0.35	35.14	39.73%
72h	lac25 lac32 (#22)	36149.33	0.28	28.48	56.55%

72h	lac25 lac32 (#24)	36510.05	0.29	28.68	60.37%
72h	lac19 (#1)	30559.33	0.25	25.36	34.15%
72h	lac19 (#2)	32338.67	0.26	26.35	34.53%

Ding SY, Liu YS, Zeng Y, Himmel ME, Baker JO, Bayer EA. (2012). How does plant cell wall nanoscale architecture correlate with enzymatic digestibility? *Science* 338(6110):1055-60.

Point 3. Loss-of-function phenotypes from knockout of the miRNA-encoding gene are not described or discussed beyond a cursory mention in the text. However, Supplemental Figure 9b and c shows interesting phenotypes of reduced cell wall thickness and an increased intensity of phloroglucinol staining in successive internodes. Discussion of these phenotypes could enrich understanding of the function of *miR408*.

Response 3: After reading your comments, we realized that loss-of-function phenotypes of *miR408* were not well discussed in detail. Therefore, we added the discussion of *miR408* in cell wall thickening in the revised manuscript as follows:

“Previous studies showed that over-expression *miR408* can promote vegetative growth, while the impaired growth was observed in *miR408* T-DNA insertion *Arabidopsis* mutant lines (Zhang *et al.*, 2013). In addition, cell wall thickening was associated with the deposition of lignin and cellulose (Watanabe *et al.*, 2015; Liu *et al.*, 2021). In our study, the knock-out of *miR408* in poplar resulted in enhanced cell wall lignification but reduced cell wall thickness. Although lignin related genes were up-regulated in the *miR408* knock-out plants from the RNA-seq data, the genes encoding activators (VND7 and SND1) (Li *et al.*, 2012; Yang *et al.*, 2013; Takata *et al.*, 2019) in the transcriptional regulatory network pathway of secondary wall synthesis were down-regulated. Moreover, the expression level of *LBD15*, a key TF that can down-regulate the expression of cellulose synthesis genes (Zhu *et al.*, 2014), was increased to a large extent in *miR408* knock-out poplars. The genetic evidence and gene expression analysis together suggested that *miR408* may possess an additional role of regulating cell wall thickening in plants.”

Zhang H, Li L. *SQUAMOSA promoter binding protein-like7* regulated microRNA408 is required for vegetative development in *Arabidopsis*. *Plant J.* 74, 98-109 (2013).

Liu C, Yu H, Rao X, Li L, Dixon RA. Abscisic acid regulates secondary cell-wall formation and lignin deposition in *Arabidopsis thaliana* through phosphorylation of NST1. *Proc. Natl. Acad. Sci. U S A* 118,e2010911118 (2021).

- Watanabe Y, *et al.* Visualization of cellulose synthases in *Arabidopsis* secondary cell walls. *Science* 350, 198-203 (2015).
- Li Q, *et al.* Splice variant of the SND1 transcription factor is a dominant negative of SND1 members and their regulation in *Populus trichocarpa*. *Proc. Natl. Acad. Sci. U S A* 109, 14699-14704 (2012).
- Yang F, *et al.* Engineering secondary cell wall deposition in plants. *Plant Biotechnol. J* 11, 325-335 (2013).
- Takata N, *et al.* *Populus* NST/SND orthologs are key regulators of secondary cell wall formation in wood fibers, phloem fibers and xylem ray parenchyma cells. *Tree Physiol.* 39, 514-525 (2019).
- Zhu L, Guo J, Zhou C, Zhu J. Ectopic expression of *LBD15* affects lateral branch development and secondary cell wall synthesis in *Arabidopsis thaliana*. *Plant Growth Regul.* 73, 111-120 (2014).

We also added some discussion on the increased intensity of phloroglucinol staining of *miR408_cr* poplars in the Results part “*miR408* targets *LAC19*, *LAC25* and *LAC32* ” as follows:

“The down-regulated DEGs in *miR408_OX* poplars was enriched in the phenylpropanoid biosynthesis pathway (Figure 4b, Supplemental Figure 12b-c), indicating that *miR408* may be involved in lignin biosynthesis. In *miR408_cr* poplar, the expression of key genes in lignin biosynthesis such as *COMT1* and *CCoAOMT* was nearly 2.3 times higher than in WT plants, and the phenylpropanoid pathway genes *PAL1/2* and *C4H* were also up-regulated by 2.5- and 2.8-fold, respectively (Supplemental Figure 12c). These results indicate that the lignin biosynthesis pathway is more active in *miR408_cr* plants, consistent with the increased intensity of phloroglucinol staining (Supplemental Figure 9d).”

Supplemental Figure 9. Cross-sections of miR408_OX and knockout poplar visualized by phloroglucinol staining.

Point 4. Throughout the results, the authors should clarify the nature of the samples measured. For example, in Figure 1i, Table 1, Supplemental Figure 4, Figure 4d, the legend states that $n = 3$, without specifying whether these represent 3 transgenic lines, 3 trees sampled from one transgenic line, or three replicate samples from a single tree.

Response 4: Thank you for your helpful suggestions. Accordingly, we have added accurate description of sample numbers, and added the description in the corresponding figure and table legend of Figure 1b-c, i, Figure 2b-c, Figure 4d, Figure 5i-l, Table 1, Supplemental Table 4, Supplemental Figure 1c-e, Supplemental Figure 4, Supplemental Figure 8b, Supplemental Figure 9a-b and d, Supplemental Figure 10, Supplemental Figure 11, Supplemental Figure 12 e-f.

Point 5: Figure 4 also indicates that Lac 47 and Lac 55 are potential targets of *miR408* but no data are presented – are there changes in the expression levels of these two laccases? Supplemental Figure 12c shows altered expression ratios of Laccases 1, 3, 4, 10, 11 and 17,

but not 19, 25 or 32 in the *miR408*-expressing lines. Is this a typographical error? Or is the expression of these other laccase genes also impacted in the laccase triple mutant?

Response 5: To address this issue, we carried out qRT-PCR and 5' RACE for the identification of target genes.

- See Response 2 to Reviewer 1.
- As for the question about other laccases in Supplemental Figure 12c, we confirm that the altered expression ratios are not a typographical error.
- To confirm the expression level of *LAC1*, *LAC3*, *LAC4*, *LAC10*, *LAC11* and *LAC17*, we searched in the RNA-seq data between *miR408_OX* and WT. The results showed their transcript levels indeed changed in the RNA-seq data.
- The transcript levels of *LAC19*, *LAC25* and *LAC32* in the original version were not shown in Supplemental Figure 12. They have been added in Supplemental Figure 12d in our revised version.
- To confirm the transcript levels of *LAC1*, *LAC3*, *LAC4*, *LAC10*, *LAC11* and *LAC17*, we searched in the RNA-seq data between the triple laccase mutants and WT (Table SS2). The results showed that the transcript levels of *LAC1*, *LAC4* and *LAC11* were downregulated, while *LAC10* was upregulated. In addition, *LAC3* and *LAC17* were almost unexpressed in stem.
- We speculate that the changes in transcript levels of *LAC1*, *LAC4*, *LAC10* and *LAC11* were probably because that there are 55 laccases in poplar, and their functions can be redundant. *LAC1*, *LAC4*, *LAC10* and *LAC11* can compensate for the loss of *LAC19*, *LAC25* and *LAC32*.

Supplemental Figure 12. Transcriptomic and qRT-PCR analyses of 3-month-old WT and *miR408_OX* and knockout plants.

Figure SS1. Agarose gel electrophoresis showing the 5' RACE products and size markers (M) are shown. (This data is not displayed in the manuscript or supplemental materials.)

Table SS2. The fpkm of laccases from the transcriptome data between *triple laccase mutants* and WT.

Gene_id	WT_1 _fpkm	WT_2 _fpkm	#4_1_f pkm	#4_2_f pkm	#14_1 _fpkm	#14_2 _fpkm	#16_1 _fpkm	#16_2 _fpkm
Potri.001G054600 (LAC1)	0.9743	1.0042	0.1943	0.7878	0.0449	0	0.2574	0
Potri.001G206200 (LAC3)	0	0	0	0	0	0	0	0
Potri.001G248700 (LAC4)	3.2011	4.1398	1.0641	1.7462	0.9855	1.0400	1.4498	1.2467
Potri.005G200700 (LAC10)	0.1502	0	0.4995	1.4202	1.8044	2.8807	0.2646	2.0631
Potri.006G087100 (LAC11)	32.306 1	31.041 6	12.653 8	17.525 5	0.4056	16.124 6	2.3207	1.2829
Potri.008G064000 (LAC17)	0.1018	0	0	0.0874	0.1410	0.1984	0	0

Point 6. Line 267 refers to Figure 4e to g, but these panels are not part of Figure 4.

Response 6: Sorry for the unclear explanation here. This has now been clarified. The sentence beginning “*In vitro* assays with luciferase reporters (Supplemental Figure 14a-b) and 5’ RACE (Figure 4e-g) confirmed that the LACs were direct targets of *miR408*” has been changed into “*In vitro* assays with luciferase reporters (Supplemental Figure 14a-b) confirmed that *miR408* can negatively regulate the expression of *LAC19*, *LAC25* and *LAC32*. Based on 5’ RACE assay of *LAC19*, the red line in exon 2 shows the *miR408*-guided cleavage site, and the two black arrows show the detailed nucleotide cleavage positions (Figure 4e). The term 7/20 means that seven of twenty clones from the PCR products contained an *miR408*-guided cleavage 5’ end that mapped precisely to exon 2. Based on 5’ RACE assays of *LAC25* (Figure 4f) and *LAC32* (Figure 4g), the cleavage sites were all located at exon 2, and six and seven, respectively from the twenty PCR products mapped precisely to the cleavage sites”.

Point 7. The authors should cite relevant literature and discuss their own findings in the context of literature with respect to the function of laccases in lignification. For example, a dirigent protein (Dirigent protein 23) is also a predicted target gene of *miR408* (Supplemental Table 2), and this class of protein has also been implicated in lignin cross-linking. How is its expression affected in the triple laccase mutant and does this impact (or not) the interpretation of the triple mutant phenotype?

Response 7: Thanks for the helpful suggestions. As suggested, we added several sentences to the Discussion section; see Response 6 to Reviewer 2.

To clarify the function of the predicted target gene, *dirigent 23* (Potri.001G214600) as suggested, we searched its transcript levels from the transcriptome data between *MIR408_OX* and WT (Table 2 in our reply), and between the triple laccase mutants and WT (Table 3 in our reply; Data not show in this paper). The results showed that *dirigent 23* (Potri.001G214600) is almost not expressed in WT, *miR408_OX* and triple laccase mutants, as shown in the following tables, suggesting that the expression level of this gene is very low in stem and therefore unlikely to play a crucial role in stem development, implying *dirigent 23* (Potri.001G214600) is not a key gene in lignin cross-linking. This is consistent with only very few reports suggesting roles for dirigent proteins in lignin (as opposed to lignan) biosynthesis, although very recent data suggest involvement of DPs in lignification in the Casparian strip.

Table 2. The fpkm of *dirigent 23* (Potri.001G214600) from the transcriptome data between *MIR408_OX* and WT.

Gene_id	WT_1 _fpkm	WT_2 _fpkm	miR408_ OX_1_1 fpkm	miR408 _OX_1_ 2 fpkm	miR408 _OX_6_ 1 fpkm	miR408_ OX_6_2 fpkm
Potri.001G214600 (dirigent 23)	0	0	0	0.1	0	0

Table 3. The fpkm of *dirigent 23* (Potri.001G214600) from the transcriptome data between *triple laccase mutants* and WT.

Gene_id	WT_1 _fpkm	WT_2 _fpkm	#4_1_f pkm	#4_2_f pkm	#14_1 _fpkm	#14_2 _fpkm	#16_1 _fpkm	#16_2 _fpkm
Potri.001G214600 (dirigent 23)	0	0	0	0	0	0	0	0

Point 8: Supplemental Figure 8a – please clarify if “cell wall residues” refers to cell walls after cellulase treatment, or if this is the total sugar content of the starting cell wall materials.

Response 8: Sorry for the unclear definition of “Cell wall residues”. Here we use the term to refer to the cell wall material extracted by methanol and chloroform after grinding through a 40-mesh sieve. The specific extraction process is as follows: cell wall residues were generated by extracting plant tissue with methanol (three times at 37°C for 1.5 h) and chloroform: methanol (2:1) (three times at 37°C for 1.5 h). The samples were then washed

three times with water at 37°C for 1.5 h and lyophilized for 48 h (Fu *et al.*, 2011; Jackson *et al.*, 2008). To make it clear, we also added the description in the Methods part “Determination of saccharification efficiency”.

Jackson, L. A., Shadle, G. L., Zhou, R., Jin, N., Chen, F., Dixon, R. A. (2008). Improving saccharification efficiency of alfalfa stems through modification of the terminal stages of monoglignol biosynthesis. *Bioenergy Research*, 1(3), 180.

Fu C, Mielenz JR, Xiao X, Ge Y, Hamilton CY, Rodriguez M Jr, Chen F, Foston M, Ragauskas A, Bouton J, Dixon RA, Wang ZY (2011) Genetic manipulation of lignin reduces recalcitrance and improves ethanol production from switchgrass. *Proc Natl Acad Sci U S A*. 108(9): 3803-8.

Point 9: Supplemental Figure 14b legend, where are the “lower panels” showing bright field photographs?

Response 9: We apologize for missing the bright field photographs after revision. Now the bright field photographs have been added as Figure S14b.

Supplemental Figure 14. Functional identification of targets of *miR408* in planta.

Reviewers' Comments:

Reviewer #1:

Remarks to the Author:

In the revised manuscript, the authors have adequately addressed all my comments.

Jin-Gui Chen

Reviewer #2:

Remarks to the Author:

The authors have addressed my concerns and questions, and recommend it to be accepted for publication.

Reviewer #3:

Remarks to the Author:

The authors have satisfactorily addressed all concerns in my previous review.

Maureen McCann

Response to Reviewers

Comments of Reviewer 1:

In the revised manuscript, the authors have adequately addressed all my comments.

Jin-Gui Chen.

Response: We appreciate your valuable suggestions in the first and second submissions which greatly improved our manuscript.

Comments of Reviewer 2:

The authors have addressed my concerns and questions, and recommend it to be accepted for publication.

Response: Many thanks for the effort to help improving our work.

Comments of Reviewer 3:

The authors have satisfactorily addressed all concerns in my previous review.

Maureen McCann

Response: Many thanks for your positive and constructive review which greatly improved our manuscript. We are pleased that our revisions were able to address all of your concerns and comments to your satisfaction.